# Dual-Kernel Adapter: Expanding Spatial Horizons for Data-Constrained Medical Image Analysis

**Ziquan Zhu**[1*], **Hanruo Zhu**[1*], **Si-Yuan Lu**[2*], **Xiang Li**[3], **Yanda Meng**[4], **Gaojie Jin**[5]
**Lu Yin**[6], **Lijie Hu**[7], **Di Wang**[7], **Lu Liu**[5], **Tianjin Huang**[5,9†]

[1] School of Computing and Mathematical Sciences, University of Leicester, Leicester, UK
[2] School of Communications and Information Engineering, Nanjing University of Posts and Telecommunications, Nanjing, China
[3] Department of Computer Science, University of Bristol, Bristol, UK
[4] Department of Bioengineering, King Abdullah University of Science and Technology, Thuwal, Saudi Arabia
[5] Department of Computer Science, University of Exeter, Exeter, UK
[6] Computer Science Research Centre, University of Surrey, Guildford, UK
[7] Machine Learning Department, Mohamed bin Zayed University of Artificial Intelligence, Abu Dhabi, UAE
[8] Division of Computer, Electrical and Mathematical Sciences and Engineering, King Abdullah University of Science and Technology, Thuwal, Saudi Arabia
[9] Department of Mathematics and Computer Science, Eindhoven University of Technology, Eindhoven, NL

## Abstract

Adapters have become a widely adopted strategy for efficient fine-tuning of large pretrained models, particularly in resource-constrained settings. However, their performance under extreme data scarcity—common in medical imaging due to high annotation costs, privacy regulations, and fragmented datasets—remains underexplored. In this work, we present the first comprehensive study of adapter-based fine-tuning for large pretrained models in low-data medical imaging scenarios. We find that, contrary to their promise, conventional Adapters can degrade performance under severe data constraints, performing even worse than simple linear probing when trained on less than 1% of the corresponding training data. Through systematic analysis, we identify a sharp reduction in Effective Receptive Field (ERF) as a key factor behind this degradation. Motivated by these findings, we propose the Dual-Kernel Adapter (`DKA`), a lightweight module that expands spatial context via large-kernel convolutions while preserving local detail with small-kernel counterparts. Extensive experiments across diverse classification and segmentation benchmarks show that `DKA` significantly outperforms existing Adapter methods, establishing new leading results in both data-constrained and data-rich regimes. Code is available at `https://github.com/misswayguy/DKA`.

## 1 Introduction

The rapid proliferation of large pretrained models has significantly advanced various fields such as natural language processing (Chowdhary & Chowdhary, 2020) and computer vision (Szeliski, 2022), yet it has also amplified challenges related to computational overhead (Thompson et al., 2020), memory consumption (Mahendran, 2021), and the complexity of downstream adaptation (Jiang et al., 2024), especially when deploying these models in specialized domains like medical imaging (Strubell et al., 2020; Ji et al., 2021; Wang et al., 2025). To address these issues, adapter-based fine-tuning (Wang et al., 2020b; Wu et al., 2025; Gong et al., 2023; Hu et al., 2024; Chen et al., 2024) has

---

*Equal contribution.
†Corresponding author: T.Huang2@exeter.ac.uk.

emerged as a popular strategy, enabling efficient adaptation by adjusting only a small subset of model parameters rather than performing full fine-tuning.

In medical imaging, assembling large, well-annotasted datasets is notoriously costly: expert radiologists must painstakingly delineate structures in high-resolution 2-D and 3-D scans, and inter-observer variability further inflates the annotation burden (Suzuki, 2017; Ritter et al., 2011). Strict privacy regulations such as HIPAA (U.S. Department of Health & Human Services, 2003) and the GDPR (European Union, 2016), coupled with heterogeneous institutional policies, severely limit data sharing, fragmenting what little data exist (Schäfer et al., 2023; Wang et al., 2021). As a result, many clinically important downstream tasks still operate in a pronounced low-data regime. This scenario naturally leads to the critical question:

*Can standard Adapter perform effectively in medical imaging tasks under constrained data?*

In this paper, we first provide a comprehensive evaluation and analysis of applying conventional Adapter (Houlsby et al., 2019) under constrained-data scenarios across various datasets and backbone architectures. Our study reveals several key insights:

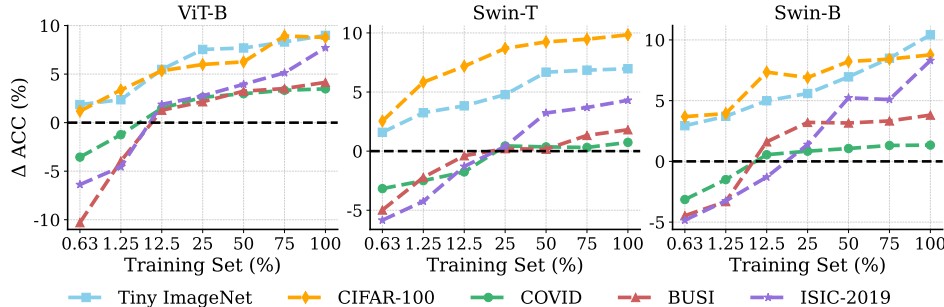

Figure 1: **Performance of Adapter across Various Training Data Sizes.** $\Delta$ACC = $\text{ACC}_{Linear Probing+Adapter} - \text{ACC}_{Linear Probing}$. Experiments are conducted on the pretrained ViT-B, Swin-T, and Swin-B backbones for Tiny ImageNet and CIFAR-100 (In-Domain), COVID, BUSI, and ISIC-2019 (Out-of-Domain).

- *Data Size Significantly Impacts Adapter Performance.* We observe that the benefits of using Adapters diminish substantially as the size of the training dataset decreases. This effect is particularly pronounced for medical imaging (see Figure 1).

- *Adapters Can Harm Performance Under Severe Low Data Settings.* When the training data size is reduced to 1% or less, specifically in the context of adapting large pretrained models to medical imaging, Adapters perform worse than linear probing. This indicates that, under extreme data constraints, Adapters may negatively affect model adaptation (see Figure 1).

- *Effective Receptive Field (ERF) Decreases with Reduced Training Data.* Visualization of the ERF demonstrates that smaller training datasets result in reduced ERF, offering a plausible explanation for the observed performance degradation (see Figure 2).

Inspired by these insights, we propose a Dual-Kernel Adapter (`DKA`) that explicitly enlarges the ERF of standard Adapter modules. Each `DKA` module introduces a dual-branch convolution design: one branch leverages a depthwise large-kernel convolution to broaden the ERF, while the other employs a depthwise small-kernel convolution to preserve fine-grained local details. We evaluate `DKA` on a range of medical imaging tasks, including both classification and segmentation, across diverse datasets and large pretrained models. Experimental results demonstrate that `DKA` consistently outperforms existing methods in both data-constrained and data-rich settings.

**Summary of Contributions**

⋆ We present the first systematic study of adapter-based fine-tuning for large pretrained models in low-data medical-imaging scenarios, showing that the conventional Adapter can actually degrade performance under severe data scarcity.

⋆ We introduce the `Dual-Kernel Adapter` (DKA), a lightweight module that pairs large- and small-receptive-field convolutions in parallel, simultaneously broadening spatial context and preserving fine-grained detail.

⋆ Extensive experiments on multiple segmentation and classification benchmarks demonstrate that `DKA` sets new state of the art, and ablation studies reveal that using asynchronous learning rates between adapters and linear head is critical to its gains.

## 2 UNDERSTANDING STANDARD ADAPTER IN CONSTRAINED-DATA SETTINGS

Adapters have gained popularity in medical image analysis due to their parameter efficiency and adaptability for fine-tuning large pretrained models. However, their performance under constrained data conditions remains underexplored. In this section, we investigate this critical aspect using a diverse set of datasets, including Tiny ImageNet (Le & Yang, 2015), CIFAR-100 (Sharma et al., 2018), COVID (Chowdhury et al., 2020), BUSI (Zhang et al., 2022), and ISIC-2019 (Gessert et al., 2020). Our experiments utilize three backbones—ViT-B (Dosovitskiy et al., 2021), Swin-T (Liu et al., 2021), and Swin-B (Liu et al., 2021)—all pretrained on the ImageNet (Deng et al., 2009), to evaluate the impact of Adapters across varying data sizes, ranging from 0.63% to 100% of the training data. We report the difference in accuracy between the application of Adapters and the non-application of Adapters, which is denoted by $\Delta$ACC = $\text{ACC}_{LinearProbing+Adapter}$ − $\text{ACC}_{LinearProbing}$. Our key observations are summarized as follows.

① **Degraded Adapter Performance with Less Training Data.** In Figure 1, we consistently observe that the performance gains provided by Adapters diminish across multiple tasks and pretrained models. Notably, this decline is significantly more pronounced in medical datasets (COVID, BUSI, ISIC-2019) compared to natural-vision datasets (TinyImageNet, CIFAR-100). A plausible explanation is that medical tasks represent out-of-domain scenarios, requiring feature representations that diverge considerably from those learned by the original pretrained models. This challenge is further exacerbated in low-data settings, where learning domain-specific features becomes more difficult. In contrast, natural-vision tasks remain largely in-domain, aligning more closely with the pretraining distribution, and thus benefiting more from adapter-based fine-tuning.

② **Negative Effects of Adapters in Extremely Low Training Data in Medical Imaging.** We further observe that in medical imaging tasks, when training data is limited to 1% or less, the performance gain from applying Adapters becomes negative, indicating a detrimental impact on model performance. Unlike natural images, medical images typically exhibit low contrast, ambiguous boundaries, and small or irregular pathological structures (Zhang et al., 2024), which usually demand a large effective receptive field (ERF) to capture long-range contextual dependencies. However, standard Adapter do not possess a strong inductive bias toward expanding the ERF. **We hypothesize** that *under limited supervision, this limitation restricts the Adapter's capacity to learn spatially dispersed features and long-range contextual dependence, thereby contributing to the observed degradation in performance.*

③ **Reduced Effective Receptive Field Under Constrained Training Data.** To validate whether reduced supervision limits the Adapter's ability to capture spatially dispersed features and long-range contextual dependencies, we visualize the ERF of Adapters trained on varying proportions of the training set, ranging from 0.63% to 100%, using the COVID dataset and the pretrained ViT-B model. Following the definition in (Araujo et al., 2019), the ERF of a neural network layer refers to the region encompassing all input pixels that exert a non-negligible influence on a given output unit. As shown in Figure 2, the ERF becomes progressively smaller as the amount of training data decreases. This observation supports our hypothesis that limited supervision restricts the Adapter's ability to learn spatially diverse patterns and long-range contextual relationships, which are particularly crucial in medical imaging tasks. We provide complementary ERF analysis on an alternative backbone (Swin-T) in Appendix F.4, which leads to consistent conclusions, reinforcing the generality of our observations.

These findings indicate that standard Adapter fail to enlarge the ERF in low-data settings, which in turn degrades model performance. Consequently, it is crucial to develop new Adapter architectures with a built-in inductive bias toward expanding the ERF.

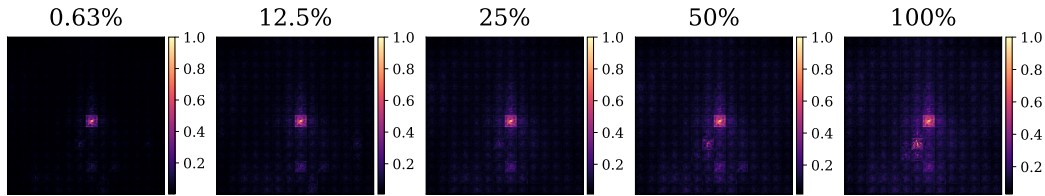

**Figure 2: Effective Receptive Field of Standard Adapters Across Varying Training Set Ratios.** Experiments are conducted on the COVID dataset using the pretrained ViT-B.

## 3 DUAL-KERNEL ADAPTER

To mitigate the adverse effects of standard Adapter in low-data regimes, we propose the `Dual-Kernel Adapter` (DKA), which explicitly integrates a large-kernel convolution to expand the ERF and a small-kernel convolution to preserve local spatial details. Prior studies have demonstrated that large kernels introduce a strong inductive bias toward capturing broader contextual information by expanding the ERF (Huang et al., 2023; Liu et al., 2022; Ding et al., 2022).

As illustrated in Figure 3, DKA first reduces the dimensionality of the input features through a linear down-projection. After this step, the patch tokens are reshaped back to their 2D spatial layout so that depthwise convolutions can be applied in the image domain. The reduced features are then processed in parallel by two depthwise convolution branches—a computationally efficient form of convolution that applies a distinct filter to each input channel (Chollet, 2017). One branch employs a large kernel to significantly enlarge the receptive field, facilitating the modeling of long-range dependencies. The other branch uses a smaller kernel to retain fine-grained spatial features essential for capturing localized structures. The outputs from the two branches are aggregated via element-wise summation, followed by a GELU activation, a linear up-projection, and a residual connection to the input.

Formally, the DKA operation can be expressed as:

$$f_{\text{DKA}}(\boldsymbol{x}) = \boldsymbol{x} + \text{Up}(\sigma(\text{DWConv}_{\text{large}}(\text{Down}(\boldsymbol{x})) + \text{DWConv}_{\text{small}}(\text{Down}(\boldsymbol{x})))) \quad (1)$$

where $\text{Down}(\cdot)$ and $\text{Up}(\cdot)$ denote linear projection layers, $\text{DWConv}_{\text{large}}$ and $\text{DWConv}_{\text{small}}$ are depthwise convolutions with kernel sizes 51 and 5 respectively, and $\sigma$ is the GELU activation.

## 4 EXPERIMENTS

To comprehensively evaluate the performance of DKA, we conduct extensive experiments on medical imaging tasks, spanning classification and segmentation benchmarks across diverse datasets, and additionally, models pretrained on both natural and medical domains.

**Pretrained Models.** We consider two representative categories of pretrained models. Natural-pretrained Models. For classification, we conduct experiments using ViT-B (Dosovitskiy et al., 2021) and Swin-B (Liu et al., 2021), both pretrained on the ImageNet. For segmentation, we adopt Segmenter-B (Strudel et al., 2021) in the OpenMMLab MMSegmentation (Contributors, 2020). Medical-pretrained Models. To assess domain adaptability, we further evaluate DKA on medical backbones, including RadImageNet-pretrained ResNet-50 (Mei et al., 2022) for classification and MedSAM (Ma et al., 2024) for segmentation.

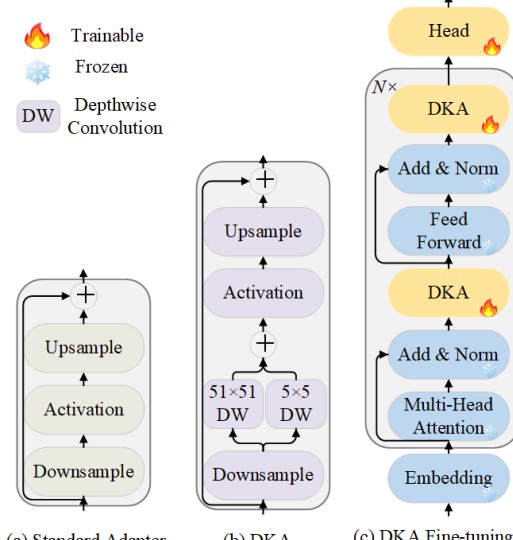

Figure 3: **Overview of the DKA Module.** (a) Standard Adapter. (b) The proposed DKA module. (c) DKA Fine-tuning.

**Datasets and Metrics.** We evaluate the proposed DKA across both medical classification and segmentation tasks to ensure broad applicability. Medical Image Classification. We utilize three widely-adopted medical image classification datasets: COVID (Chowdhury et al., 2020), BUSI (Zhang et al., 2022), and ISIC-2019 (Gessert et al., 2020). The COVID dataset comprises chest X-ray images for COVID-19 diagnosis. The BUSI dataset includes breast ultrasound images categorized as benign, malignant, or normal. The ISIC-2019 dataset is a large-scale dermoscopic dataset designed for multi-class skin lesion classification. For each dataset, we report *Top-1 Accuracy (ACC)*, *F1 Score (F1)*; and *Sensitivity (SEN)* under varying proportions of training data. Medical Image Segmentation. We further assess DKA on three representative segmentation datasets: BRATS (Menze et al., 2014), BUSI (Zhang et al., 2022), and ISIC-2018 (Codella et al., 2019). BRATS focuses on brain tumor segmentation using multi-modal MRI scans. BUSI provides breast ultrasound images with annotated tumor regions. ISIC-2018 contains dermoscopic images with pixel-level annotations for skin lesion segmentation. We evaluate segmentation performance using standard metrics: *mean Intersection over Union (mIoU)* and *Dice coefficient (Dice)*. More details are provided in Appendix B.1.

**Baselines.** We compare DKA against a broad range of baselines, categorized into three groups: Standard Fine-tuning Methods: (1) *Linear Probing*: freezing the backbone and tuning only the head; (2) *Full Fine-tuning*: updating all model parameters during downstream training. Adapter-tuning Methods: (1) *Adapter (Houlsby et al., 2019)*: adding adapters within each transformer block; (2) *AdapterFormer (Chen et al., 2022)*: adding parallel adapters with learnable scaling to each MLP layer; (3) *CIAT (Zhu et al., 2021)*: adding adapters before transformer blocks and parallel adapters within blocks; (4) *Convpass (Jie et al., 2024)*: enhancing adapters with parallel $3 \times 3$ convolution branches; (5) *AIM (Yang et al., 2023)*: combining sequential and parallel adapters across spatial and temporal attention pathways. Other Parameter-efficient Fine-tuning (PEFT) Methods: (1) *Prompt Tuning (Jia et al., 2022)*: fine-tuning extra learnable tokens; (2) *LoRA (Hu et al., 2022)*: injecting low-rank trainable matrices into attention modules; (3) *BitFit (Zaken et al., 2021)*:fine-tuning only the bias terms of pretrained models.

**Implementation Details.** Following the common training protocol, we freeze all pretrained model weights and update only the parameters of DKA and head during fine-tuning. Specifically, DKA modules are inserted within transformer blocks following the placement strategy described in (Yin et al., 2024). For the DKA module, the middle dimension $\hat{d}$ is set to 16 for classification tasks and 192 for segmentation tasks, as discussed in Section 4.5. The learning rates are set to 1e-4 for the task head and 1e-3 for the DKA modules, as shown in Section 4.4. Classification models are trained for 100 epochs, while segmentation models are trained for 300 epochs, balancing the need for convergence across task complexities. For experiments under constrained supervision (i.e., training with less than 100% of the training set), we perform a 5-fold cross-validation on the training set while keeping the test set fixed. All reported results are averaged across the cross-validation folds. More details can be found in Appendix B.2.

Table 1: **Comparison of Baselines and DKA on Three Classification Datasets Across Varying Data Sizes.** Results are reported in terms of ACC (%). Experiments are based on the pretrained ViT-B. The best results are highlighted in bold.

| Methods | COVID | | | BUSI | | | ISIC-2019 | | |
|---|---|---|---|---|---|---|---|---|---|
| | 0.63% | 1.25% | 100% | 0.63% | 1.25% | 100% | 0.63% | 1.25% | 100% |
| Full Fine-tuning | 87.43 | 88.00 | 98.43 | 71.17 | 76.73 | 94.62 | 60.04 | 61.21 | 82.05 |
| Linear Probing | 86.84 | 87.50 | 94.85 | 73.48 | 77.64 | 89.78 | 59.15 | 59.44 | 71.83 |
| BitFit (Zaken et al., 2021) | 73.91 | 79.65 | 96.95 | 57.19 | 60.20 | 88.27 | 50.80 | 53.22 | 79.84 |
| Prompt (Jia et al., 2022) | 77.91 | 83.75 | 98.45 | 61.34 | 64.30 | 93.07 | 52.55 | 53.59 | 81.02 |
| LoRA (Hu et al., 2022) | 80.43 | 85.91 | 98.73 | 63.64 | 67.41 | 94.75 | 51.08 | 53.96 | 81.75 |
| Adapter (Houlsby et al., 2019) | 83.29 | 86.26 | 98.33 | 63.18 | 73.68 | 93.33 | 52.77 | 54.88 | 79.54 |
| Adapterformer (Chen et al., 2022) | 82.46 | 84.32 | 98.18 | 63.42 | 72.75 | 92.65 | 51.62 | 52.71 | 78.19 |
| Convpass (Jie et al., 2024) | 84.72 | 86.94 | 98.45 | 64.83 | 74.63 | 93.97 | 54.72 | 56.25 | 80.45 |
| CIAT (Zhu et al., 2021) | 77.34 | 82.85 | 96.54 | 60.28 | 65.07 | 89.86 | 48.35 | 49.96 | 72.18 |
| AIM (Yang et al., 2023) | 80.92 | 83.55 | 97.23 | 62.72 | 70.34 | 90.12 | 50.12 | 52.72 | 77.39 |
| **DKA** | **89.01** | **91.06** | **99.21** | **74.23** | **79.46** | **95.89** | **60.52** | **62.32** | **83.09** |

Table 2: **Comparison of Baselines and DKA on Three Segmentation Datasets Under Varying Data Sizes.** Experiments are based on the pretrained Segmenter-B. mIoU is reported as percentages. The best results are highlighted in bold.

| Methods | BRATS | | | BUSI | | | ISIC-2018 | | |
|---|---|---|---|---|---|---|---|---|---|
| | 0.63% | 1.25% | 100% | 0.63% | 1.25% | 100% | 0.63% | 1.25% | 100% |
| Full Fine-tuning | 9.25 | 22.39 | 73.08 | 26.67 | 32.31 | 57.41 | 62.27 | 73.63 | 77.58 |
| Linear Probing | 7.95 | 20.20 | 69.86 | 25.53 | 31.96 | 54.07 | 60.90 | 71.06 | 74.10 |
| BitFit (Zaken et al., 2021) | 1.20 | 14.33 | 63.52 | 7.13 | 17.08 | 52.56 | 53.21 | 65.84 | 73.10 |
| Prompt (Jia et al., 2022) | 1.22 | 15.21 | 64.53 | 9.19 | 18.77 | 53.57 | 56.56 | 67.40 | 73.61 |
| LoRA (Hu et al., 2022) | 3.84 | 16.19 | 68.48 | 14.10 | 22.39 | 53.96 | 58.70 | 69.03 | 73.84 |
| Adapter (Houlsby et al., 2019) | 6.16 | 18.95 | 72.02 | 18.18 | 25.90 | 55.01 | 59.78 | 72.80 | 76.71 |
| Adapterformer (Chen et al., 2022) | 5.99 | 18.77 | 72.54 | 17.41 | 25.68 | 55.14 | 59.67 | 72.85 | 76.58 |
| Convpass (Jie et al., 2024) | 7.13 | 19.64 | 73.32 | 19.84 | 28.52 | 56.09 | 60.19 | 73.56 | 77.54 |
| CIAT (Zhu et al., 2021) | 3.80 | 16.81 | 70.41 | 13.40 | 20.20 | 54.76 | 58.26 | 69.52 | 75.09 |
| AIM (Yang et al., 2023) | 4.58 | 17.61 | 71.98 | 15.40 | 23.54 | 54.78 | 59.24 | 72.05 | 76.65 |
| **DKA** | **9.47** | **23.02** | **74.96** | **26.85** | **34.52** | **58.90** | **63.13** | **74.27** | **78.53** |

## 4.1 SUPERIOR PERFORMANCE ON CONSTRAINED DATA

We evaluate DKA under constrained-data settings (0.63% and 1.25%) and the full-data setting. Experiments are organized by the type of pretrained backbone: natural-image models (ViT-B, Segmenter-B) and medical-image models (RadImageNet-ResNet-50, MedSAM). This design enables a clear examination of DKA's generalization across different pretraining sources.

**Natural-pretrained Models.** We first evaluate DKA using natural-pretrained backbones, as summarized in Tables 1 and 2, DKA consistently outperforms all baselines across classification and segmentation tasks. Notably, under low-data regimes (0.63% and 1.25%), DKA even surpasses full fine-tuning and linear probing, while other PEFT methods exhibit clear performance degradation. These observations confirm the effectiveness of DKA when adapting natural-pretrained models to medical domains. Additional results on the natural-pretrained ViT-B and Segmenter-B are provided in Appendix D and C. Furthermore, similar performance improvements are observed when DKA is applied to other natural-pretrained models such as Swin-B (see Appendix F.1 for details).

**Medical-pretrained Models.** We further evaluate DKA on medical-pretrained backbones, including RadImageNet-pretrained ResNet-50 (Mei et al., 2022) for classification and MedSAM (Ma et al., 2024) for segmentation. As reported in Table 3, DKA consistently surpasses full fine-tuning, linear probing, and other PEFT baselines across ISIC-2019 and BUSI under all data scales. These improvements demonstrate that the gains of DKA are not limited to natural image pretraining, but also generalize effectively to domain-specific medical large pretrained models. More results are reported in Appendix E.

Table 3: **Performance of DKA with medical-pretrained models.** (a) Classification results on the ISIC-2019 using RadImageNet-pretrained ResNet-50 by reporting ACC (%). (b) Segmentation results on the BUSI using MedSAM by reporting mIoU (%). The best results are highlighted in bold.

(a) Classification Results.

| Methods | 0.63% | 1.25% | 100% |
|---|---|---|---|
| Full Fine-tuning | 52.70 | 55.17 | 76.63 |
| Linear Probing | 51.27 | 53.71 | 67.39 |
| BitFit | 48.84 | 51.62 | 70.38 |
| Prompt | 50.12 | 53.29 | 73.31 |
| LoRA | 50.03 | 52.51 | 72.92 |
| Adapter | 51.32 | 54.04 | 74.26 |
| DKA | **53.69** | **56.58** | **78.56** |

(b) Segmentation Results.

| Methods | 0.63% | 1.25% | 100% |
|---|---|---|---|
| Full Fine-tuning | 36.21 | 45.04 | 70.62 |
| Linear Probing | 34.72 | 42.76 | 66.54 |
| BitFit | 27.85 | 36.92 | 64.43 |
| Prompt | 29.57 | 38.71 | 64.62 |
| LoRA | 32.39 | 40.35 | 67.04 |
| Adapter | 35.40 | 43.62 | 68.06 |
| DKA | **37.13** | **46.27** | **72.53** |

## 4.2 ERF VISUALIZATION

To assess whether `DKA` effectively expands the effective receptive field (ERF), we visualize the ERFs of `DKA` alongside those of other adapter-tuning baselines, including Adapter (Houlsby et al., 2019), AdapterFormer (Chen et al., 2022), Convpass (Jie et al., 2024), CIAT (Zhu et al., 2021), and AIM (Yang et al., 2023), under both the constrained (0.63%) and the full (100%) training data settings. As shown in Figure 4, `DKA` consistently exhibits the broadest ERF across both settings, demonstrating its superior ability to capture extensive spatial context even under constrained-data conditions. In contrast, other baselines yield more localized ERFs, particularly in the constrained-data setting, which likely contributes to their comparatively weaker performance.

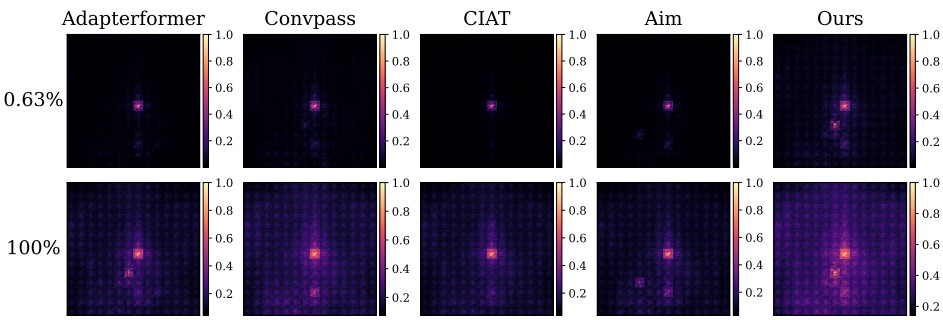

Figure 4: **Effective Receptive Field of `DKA` and Other Adapter-based Methods.** Experiments are conducted on the COVID dataset based on the pretrained ViT-B under both constrained (0.63%) and full (100%) training settings.

## 4.3 LARGE KERNEL MATTERS INSTEAD OF TRAINABLE PARAMETERS

Given that `DKA` introduces a slightly higher number of trainable parameters, a natural question arises: *Are the observed performance gains primarily attributable to the increased parameter count or to the effect of the large kernel?* To investigate this, we perform a controlled comparison where the increase in trainable parameters arises either from enlarging the kernel size or from expanding the intermediate dimension $\hat{d}$. Specifically, we adjust the middle dimension $\hat{d}$ of other adapter-based baselines so that their total trainable parameter counts align with that of `DKA` with increasing kernel sizes. We fix the middle dimension $\hat{d} = 16$ in `DKA` throughout all classification experiments, ensuring that any parameter increase stems solely from the enlarged kernel. As shown in Figure 5, the results reveal that: ❶ `DKA` consistently outperforms all baselines across the 0.63%, 1.25%, and 100% training data settings under similar parameter constraints; ❷ the performance improvements resulting from increasing the kernel size (from 11 to 51) are significantly steeper than those obtained by merely enlarging the hidden dimension, highlighting the effectiveness of large kernels in enhancing `DKA`'s representational capacity.

## 4.4 ASYNCHRONOUS LEARNING RATES MATTER

In standard adapter-based fine-tuning, it is common practice to use the same learning rate for both the adapter and the head. However, given their different roles and characteristics, this one-size-fits-all strategy may not be optimal. To investigate the effect of asynchronous learning rates for the adapter and head, we conduct experiments on the COVID dataset using a pretrained ViT-B. We systematically vary the learning rates assigned to the adapter and head, and assess the results under both the 0.63% and 1.25% training data. As shown in Figure 6, asynchronous learning rate schedules-where the adapter and head use different learning rates—often outperform symmetric configurations. We observe that the best results are not achieved when both components share the same learning rate, suggesting that the adapter and head benefit from distinct optimization dynamics. This trend holds across data scales, highlighting the importance of tuning these components independently. More results are available in Appendix F.2.

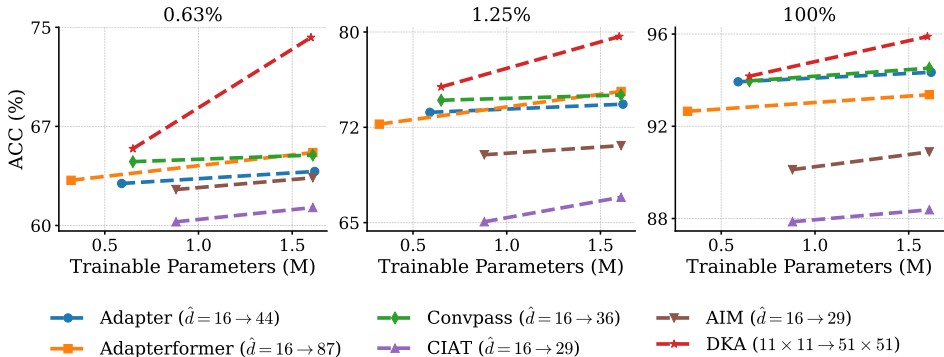

Figure 5: **Comparison of Baselines and DKA with Comparable Numbers of Trainable Parameters.** Experiments are conducted on 0.63%, 1.25%, and 100% subsets of the BUSI dataset using the pretrained ViT-B. The symbol $\hat{d}$ denotes the intermediate dimensionality in adapter-based methods.

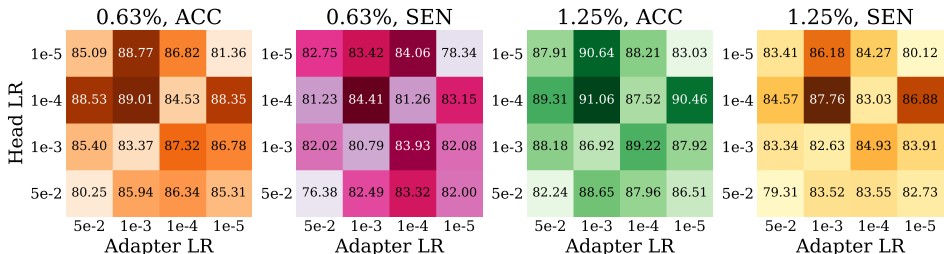

Figure 6: **Performance Comparison Across Varying Learning Rates for DKA and Classification Head.** Experiments are conducted on the COVID dataset using the pretrained ViT-B under 0.63% and 1.25% training data. Results are reported in terms of ACC (%) and SEN (%).

## 4.5 ABLATION STUDY

**Kernel Size Selection.** We investigate the impact of different kernel size combinations in DKA's dual-branch convolution design. Specifically, we sweep over five candidate sizes (3×3, 5×5, 7×7, 9×9, 11×11, 31×31, 51×51, 71×71) for both small- and large-depthwise convolution branches. As shown in Figure 7, the combination of 5×5 (small) and 51×51 (large) consistently yields the best accuracy on both the data-constrained (0.63% and 1.25%) setting and the full data setting (100%). Notably, kernel sizes that are too small or too large result in performance degradation, particularly under low-data conditions. These results support our dual-branch design choice, balancing fine-grained detail extraction and large receptive field expanding. Extra experimental results are included in Appendix F.5.

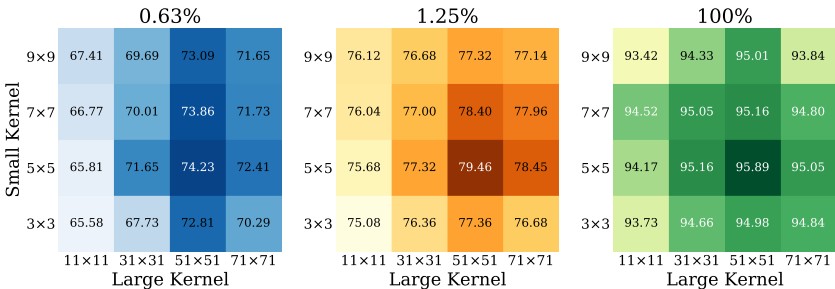

Figure 7: **Performance of Different Kernel Size Combination.** Experiments are conducted on the BUSI dataset using the pretrained ViT-B across three training setting (0.63%, 1.25%, and 100%). ACC (%) is reported.

Table 4: **Performance Comparison of Different Middle Dimensions $\hat{d}$ in DKA.** (a) Classification performance using the pretrained ViT-B on the BUSI dataset by reporting ACC (%). (b) Segmentation performance using the pretrained Segmenter-B on the ISIC-2018 dataset by reporting mIoU (%).



(a) Middle Dimension for Classification.

| $\hat{d}$ | 0.63% | 1.25% | 100% |
|---|---|---|---|
| 1 | 65.02 | 70.69 | 87.86 |
| 4 | 69.65 | 74.38 | 91.42 |
| 8 | 72.18 | 76.68 | 93.61 |
| 16 | **74.23** | **79.64** | **95.89** |
| 32 | 73.75 | 79.23 | 95.29 |

(b) Middle Dimension for Segmentation.

| $\hat{d}$ | 0.63% | 1.25% | 100% |
|---|---|---|---|
| 64 | 57.93 | 68.68 | 74.10 |
| 96 | 60.25 | 71.95 | 76.18 |
| 128 | 62.30 | 73.56 | 77.48 |
| 192 | **63.13** | **74.27** | **78.06** |
| 256 | 62.72 | 73.88 | 77.95 |



**Single vs. Dual Convolutions.** To evaluate the effectiveness of using both large- and small-depthwise convolutions in DKA, we compare the full dual-branch design ($5\times5 + 51\times51$) against single-branch variants that use only one of the two kernels. Experiments are conducted on the COVID, BUSI, and ISIC-2019 datasets across varying training set sizes. As shown in Figure 8, the dual-branch design consistently outperforms both single-branch variants, particularly in low-data regimes. While the $5\times5$ branch better captures localized detail and the $51\times51$ branch improves global coverage, neither alone matches the full-dual structure. See Appendix F.6 for additional results.

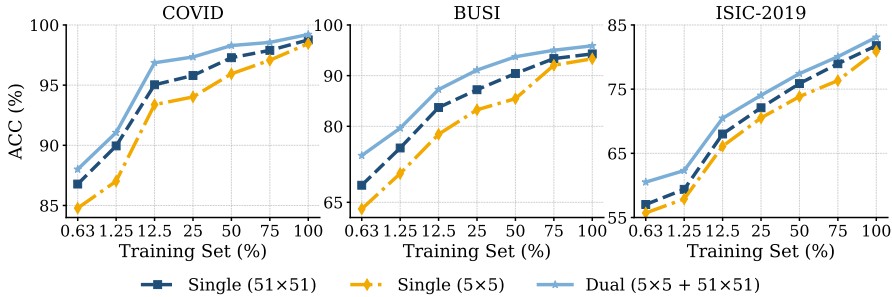

Figure 8: **Ablation for Dual-Convolution Design.** Experiments are based on the pretrained ViT-B across three classification datasets. ACC (%) is reported.

**Middle Dimension.** We investigate the impact of the middle dimension $\hat{d}$ in the DKA module for both classification and segmentation tasks. As shown in Table 4, increasing $\hat{d}$ generally leads to improved performance. For classification on the BUSI dataset, performance steadily improves from $\hat{d} = 1$ to $\hat{d} = 16$, reaching a peak at $\hat{d} = 16$, after which performance slightly declines or saturates, indicating potential overfitting or redundancy. For segmentation on the ISIC-2018 dataset, the optimal middle dimension is also moderate, with $\hat{d} = 192$ offering the best trade-off between parameter efficiency and accuracy. This observation aligns with the findings in (Chen et al., 2022), which also highlight the task-specific nature of optimal middle dimensions, emphasizing that different tasks require different parameter settings for optimal performance.

**Learnable Kernel Sizes.** To further investigate the learnable kernel sizes, following the design of Selective Kernel Networks Li et al. (2019), we implemented a learnable kernel-selection variant of our DKA. Specifically, we constructed two depthwise branches with different kernel sizes and fused them using a lightweight attention gate (global average pooling $\rightarrow$ two-layer MLP $\rightarrow$ softmax weights) that dynamically selects between the local and global branches for each input. This enables input-adaptive kernel aggregation. The results are reported in Table 5. While the learnable-kernel variant achieves competitive performance, our fixed $51 \times 51 + 5 \times 5$ design remains slightly more robust in low-data regimes (0.63% and 1.25%), likely because the learned gating requires additional data to generalize reliably. Under full data (100%), both approaches perform similarly. Overall, adaptive kernels are feasible, but our chosen configuration offers greater robustness and reliability in the low-data medical scenarios targeted in this work.

Table 5: **Comparison of Learnable Kernel Sizes and Our dDesign.** Experiments are conducted on the COVID dataset using the pretrained ViT-B.

| Methods | 0.63% | 1.25% | 100% |
|---|---|---|---|
| Learnable kernel sizes | 88.98 | 91.05 | 99.21 |
| $51 \times 51 + 5 \times 5$ (Ours) | 89.01 | 91.06 | 99.21 |

## 5 RELATED WORK

**Adapter-based Fine-tuning** adapts large pretrained models to downstream tasks by inserting and training lightweight modules while keeping the original model parameters frozen, offering significant computational and memory advantages over full fine-tuning (Xu et al., 2023; Ding et al., 2023; Han et al., 2024b). Early Adapter methods introduced task-specific bottleneck layers between transformer blocks (Houlsby et al., 2019), with subsequent innovations improving architectural designs (Wang et al., 2020b; Zhang et al., 2021) and multi-modal applications (Sung et al., 2022; Pan et al., 2022; Gao et al., 2024). Recent medical imaging adaptations demonstrate how Adapter modules can effectively transfer pretrained knowledge to diagnostic tasks while maintaining model integrity (Chen et al., 2023; Dutt et al., 2023b; Lian et al., 2024). Recent large-kernel designs such as LKA (Zhu et al., 2025) enhance global receptive fields within convolutional backbones However, adapter-based methods typically assume access to a moderate amount of labeled data (Dutt et al., 2023a; Liu et al., 2024). Their effectiveness in data-constrained settings remains underexplored, raising critical questions about their performance in such scenarios.

**Limited Data in Medical Imaging** remains a major challenge, as labeled samples are scarce due to privacy, high annotation cost, and limited experts (Shaikhina & Khovanova, 2017; Chlap et al., 2021). This constraint is especially critical in clinical practice, where collecting large and diverse datasets is difficult (Ewing, 2017; Lee & Yoon, 2017; Li et al., 2010). To address scarcity, strategies include data augmentation (Garcea et al., 2023; Goceri, 2023; Zhao et al., 2019; Islam et al., 2024), transfer learning (Raghu et al., 2019; Kim et al., 2022; Kora et al., 2022), semi-supervised learning (Huynh et al., 2022; Chebli et al., 2018; Wang et al., 2020a; Han et al., 2024a; Zhou et al., 2019), and self-supervised representation learning (Ericsson et al., 2022; Jiao et al., 2020; Ye et al., 2024; Krishnan et al., 2022). These approaches leverage unlabeled data or external sources to improve generalization under low-resource settings (Zheng et al., 2024; Lin et al., 2021), and recent work highlights large pretrained models to further reduce annotation needs (Moor et al., 2023; Zhang & Metaxas, 2024; Khan et al., 2025). Nevertheless, most rely on full fine-tuning or moderately sized datasets (Davila et al., 2024; Khan & Fang, 2023), while their integration with adapter-based methods under severe scarcity remains underexplored but highly relevant in practice.

## 6 CONCLUSION

In this paper, we revisit adapter-based fine-tuning for medical image analysis and uncovered key limitations in low-data settings: existing Adapters often struggle to capture relevant features under data scarcity, partly due to their constrained receptive field. To address this, we introduced DKA, a dual-branch adapter module that integrates large- and small-kernel depthwise convolutions to enhance the receptive field while preserving local detail. Experiments on both medical image classification and segmentation tasks, evaluated across various datasets and backbones show that DKA consistently outperform both full fine-tuning and other PEFT baselines by a good margin, particularly in the constrained-data setting, without significantly increasing parameter count.

## 7 ACKNOWLEDGMENTS

The authors acknowledge the use of resources provided by the UKRI SLAIDER project, the MRC SLAIDER-QA project, the Isambard-AI National AI Research Resource (AIRR), and the Dutch national e-infrastructure, supported by the SURF Cooperative (Project EINF-17091). Isambard-AI is operated by the University of Bristol and funded by the UK Government's Department for Science, Innovation and Technology (DSIT) via UK Research and Innovation and the Science and Technology

Facilities Council [ST/AIRR/I-A-I/1023]. Finally, we thank the anonymous reviewers for their insightful comments, which significantly improved the quality of this paper.

## 8 ETHICS STATEMENT

We confirm that this study adheres to the ICLR Code of Ethics, in every respect. All datasets used are publicly available and have been widely adopted in prior research.

## 9 REPRODUCIBILITY STATEMENT

We provide complete implementation details, including dataset splits, training settings, and evaluation protocols in Section 4, Appendix B.1, and Appendix B.2. Code and data will be released to facilitate reproducibility and further research.

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

APPENDIX

## A  THE USE OF LARGE LANGUAGE MODELS (LLMs)

Large language models (LLMs) were not used as part of the core methodology, experiments, or original research contributions. They were only used as an assistive tool for language editing and formatting.

## B  EXPERIMENT SETTINGS

### B.1  DATASETS

#### B.1.1  CLASSIFICATION DATASETS

**COVID** (Chowdhury et al., 2020): The COVID-19 Radiography Database, developed through a collaborative effort involving researchers from Qatar University, the University of Dhaka, and medical experts from Pakistan and Malaysia, includes approximately 3616 COVID-19 cases, 10,192 normal cases, 6012 lung opacity cases (representing non-COVID lung infections), and 1345 viral pneumonia cases. This dataset provides a comprehensive collection of chest X-ray (CXR) images for the diagnosis of COVID-19 and other lung conditions.

**BUSI** (Zhang et al., 2022): The BUSI (Breast Ultrasound Images) dataset, collected in 2018, comprises approximately 780 ultrasound images from 600 female patients aged 25 to 75. It includes around 437 normal, 210 benign, and 133 malignant breast lesion images, each with an average resolution of approximately 500×500 pixels. The dataset is organized into three primary categories based on the clinical classification of breast lesions: normal, benign, and malignant.

**ISIC-2019** (Gessert et al., 2020): The ISIC-2019 dataset, part of the annual ISIC (International Skin Imaging Collaboration) challenges, includes 25,331 dermoscopic images compiled from previous ISIC challenges (2017 and 2018). It spans nine diagnostic categories: melanoma, melanocytic nevus, basal cell carcinoma, actinic keratosis, benign keratosis (including solar lentigo, seborrheic keratosis, and lichen planus-like keratosis), dermatofibroma, vascular lesion, squamous cell carcinoma, and a category for images that do not fit into any of the other classes, providing a comprehensive benchmark for multi-class skin lesion classification.

#### B.1.2  SEGMENTATION DATASETS

**BRATS** (Menze et al., 2014): The BRATS (Brain Tumor Segmentation) dataset provides multi-institutional, clinically acquired multi-parametric MRI (mpMRI) scans of gliomas, including T1, post-contrast T1-weighted (T1Gd), T2, and T2-FLAIR volumes. It includes pathologically confirmed cases with expert-annotated tumor sub-regions, including the enhancing tumor (ET), tumor core (TC), and whole tumor (WT), providing a comprehensive benchmark for brain tumor segmentation.

**BUSI** (Zhang et al., 2022): The BUSI (Breast Ultrasound Images) dataset includes ground truth masks for precise lesion segmentation, facilitating the evaluation of automated lesion detection and boundary delineation methods. These masks correspond to the original ultrasound images, capturing the exact regions of interest for each lesion type. The dataset primarily supports the segmentation of benign and malignant breast lesions, providing a detailed representation of lesion morphology.

**ISIC-2018** (Codella et al., 2019): The ISIC-2018 dataset, released as part of the ISIC 2018 Task 1 challenge, contains a total of 3,694 dermoscopic images, each paired with a binary segmentation mask outlining the precise lesion boundaries. This dataset serves as a critical benchmark for evaluating automated skin lesion segmentation methods.

### B.2  IMPLEMENTATION DETAILS

Our experiments are conducted on NVIDIA RTX 3090 GPUs. The code is based on PyTorch (Paszke, 2019). Due to hardware memory constraints, we use different batch sizes for classification and segmentation tasks. Specifically, classification tasks use a batch size of 8, while segmentation tasks are restricted to a batch size of 1, as larger batch sizes lead to out-of-memory errors. The test set is fixed at 20% of the total dataset, ensuring consistent evaluation across all experimental settings. We employ Adam optimizer (Kingma, 2014) for both the head and `DKA` modules, albeit with different

learning rates for each. When reducing the training data sizes, we proportionally decrease samples per class to maintain balance. Even in extreme low-data scenarios, we ensure that each class retains some images to prevent complete absence of any category. The weights of the down-projection layers and the biases and weights of the up-projection layers are initialized to zero, providing a stable starting point for training.

# C ADDITIONAL SEGMENTATION RESULTS ON NATURAL-PRETRAINED MODELS

## C.1 SEGMENTATION RESULTS WITH ADDITIONAL METRICS

To complement the segmentation results in the main text, we provide additional evaluations in Table 6, where performance is reported in terms of Dice scores across three datasets (BRATS, BUSI, ISIC-2018) under different training ratios (0.63%, 1.25%, 100%). The results confirm that DKA consistently achieves higher Dice scores than all baseline methods across datasets and data scales, further demonstrating its robustness in capturing both local details and global context for medical image segmentation.

Table 6: **Comparison of Baselines and DKA on Three Segmentation Datasets Under Varying Data Sizes.** Experiments are based on the pretrained Segmenter-B. Results are reported in terms of Dice (%). The best results are highlighted in bold.

| Methods | BRATS | | | BUSI | | | ISIC-2018 | | |
|---|---|---|---|---|---|---|---|---|---|
| | 0.63% | 1.25% | 100% | 0.63% | 1.25% | 100% | 0.63% | 1.25% | 100% |
| Full Fine-tuning | 16.94 | 36.59 | 84.45 | 42.11 | 48.83 | 72.94 | 76.75 | 84.81 | 87.37 |
| Linear Probing | 14.72 | 33.62 | 82.26 | 40.67 | 48.44 | 70.19 | 75.70 | 83.08 | 85.12 |
| BitFit (Zaken et al., 2021) | 2.40 | 25.06 | 77.69 | 13.31 | 29.18 | 68.90 | 69.46 | 79.41 | 84.46 |
| Prompt (Jia et al., 2022) | 2.42 | 26.41 | 78.44 | 16.84 | 31.61 | 69.76 | 72.25 | 80.52 | 84.80 |
| LoRA (Hu et al., 2022) | 7.40 | 28.86 | 81.29 | 24.72 | 36.59 | 70.09 | 73.97 | 81.68 | 84.95 |
| Adapter (Houlsby et al., 2019) | 11.60 | 31.86 | 83.73 | 30.77 | 41.14 | 70.98 | 74.83 | 84.26 | 86.82 |
| Adapterformer (Chen et al., 2022) | 11.30 | 31.61 | 84.08 | 29.66 | 40.86 | 71.08 | 74.74 | 84.29 | 86.73 |
| Convpass (Jie et al., 2024) | 13.31 | 32.83 | 84.60 | 33.11 | 44.39 | 71.87 | 75.14 | 84.77 | 87.35 |
| CIAT (Zhu et al., 2021) | 7.33 | 28.72 | 82.64 | 23.63 | 33.62 | 70.77 | 73.63 | 82.02 | 85.77 |
| AIM (Yang et al., 2023) | 8.75 | 30.49 | 83.71 | 26.69 | 38.11 | 70.79 | 74.41 | 83.75 | 86.78 |
| **DKA** | **17.29** | **37.43** | **85.69** | **42.34** | **51.33** | **74.13** | **77.39** | **85.24** | **87.97** |

## C.2 SEGMENTATION RESULTS WITH MORE DATA SCALES

To further validate the performance of DKA in segmentation tasks, we present additional results across a wider range of data scales (12.5% to 75%) in Table 7. These results, covering three segmentation medical imaging datasets, provide a comprehensive assessment of our model's performance in mid-scale training regimes, complementing the extreme low-data (0.63% and 1.25%) and full-data (100%) findings discussed in Section 4.1. Consistent with our previous observations, DKA outperforms all baselines across all datasets and training set ratios, achieving the highest mIoU and Dice metrics. This demonstrates the effectiveness of DKA, highlighting its ability to effectively capture both local detail and global context across varying supervision levels, providing a reliable solution for medical image segmentation.

Table 7: **Additional Segmentation Results on Three Datasets with Varying Training Set Ratios.** Experiments are based on the pretrained Segmenter-B. Results are reported as percentages for mIoU and Dice. The best results are highlighted in bold.

| Methods | BRATS | | | | BUSI | | | | ISIC-2018 | | | |
|---|---|---|---|---|---|---|---|---|---|---|---|---|
| | 12.5% | 25% | 50% | 75% | 12.5% | 25% | 50% | 75% | 12.5% | 25% | 50% | 75% |
| mIoU (%) | | | | | | | | | | | | |
| Full Fine-tuning | 55.28 | 61.35 | 70.43 | 71.50 | 51.44 | 53.70 | 55.35 | 56.8 | 74.60 | 75.36 | 75.50 | 77.10 |
| Linear Probing | 53.22 | 59.43 | 67.75 | 68.11 | 48.93 | 50.76 | 51.97 | 53.41 | 71.65 | 71.99 | 72.18 | 73.63 |
| BitFit (Zaken et al., 2021) | 47.21 | 53.46 | 60.30 | 61.59 | 36.30 | 41.07 | 42.45 | 48.09 | 65.27 | 66.36 | 68.00 | 71.55 |
| Prompt (Jia et al., 2022) | 48.38 | 54.08 | 62.73 | 63.01 | 38.49 | 43.38 | 46.91 | 50.30 | 68.54 | 69.16 | 69.63 | 72.31 |
| LoRA (Hu et al., 2022) | 50.77 | 57.19 | 64.80 | 67.75 | 41.22 | 47.72 | 50.42 | 52.97 | 69.64 | 70.83 | 71.61 | 72.95 |
| Adapter (Houlsby et al., 2019) | 52.52 | 59.90 | 69.22 | 70.65 | 45.25 | 49.66 | 52.11 | 54.19 | 73.54 | 73.87 | 74.48 | 76.29 |
| Adapterformer (Chen et al., 2022) | 52.12 | 59.36 | 69.11 | 70.37 | 45.22 | 49.25 | 52.40 | 54.44 | 73.45 | 73.79 | 74.15 | 75.66 |
| Convpass (Jie et al., 2024) | 54.08 | 61.08 | 69.67 | 71.26 | 49.30 | 51.77 | 53.61 | 55.27 | 74.31 | 74.98 | 75.13 | 76.94 |
| CIAT (Zhu et al., 2021) | 51.70 | 58.23 | 67.59 | 68.45 | 40.74 | 47.72 | 51.71 | 53.85 | 70.83 | 71.42 | 72.34 | 74.57 |
| AIM (Yang et al., 2023) | 51.09 | 58.43 | 68.11 | 69.34 | 43.44 | 48.15 | 51.20 | 53.96 | 72.71 | 73.36 | 73.55 | 75.30 |
| **DKA** | **56.54** | **62.92** | **71.95** | **72.92** | **52.30** | **54.76** | **56.43** | **58.08** | **75.29** | **75.83** | **76.02** | **78.06** |
| Dice (%) | | | | | | | | | | | | |
| Full Fine-tuning | 71.20 | 76.05 | 82.65 | 83.38 | 67.93 | 69.88 | 71.26 | 72.45 | 85.46 | 85.95 | 86.04 | 87.07 |
| Linear Probing | 69.47 | 73.33 | 80.77 | 81.03 | 65.71 | 67.34 | 68.39 | 69.63 | 83.48 | 83.71 | 83.85 | 84.81 |
| BitFit (Zaken et al., 2021) | 64.86 | 69.43 | 76.86 | 76.14 | 53.57 | 58.29 | 61.45 | 66.42 | 78.20 | 79.21 | 80.13 | 81.78 |
| Prompt (Jia et al., 2022) | 65.21 | 70.20 | 77.10 | 77.31 | 55.59 | 60.51 | 63.86 | 67.07 | 81.33 | 81.77 | 82.10 | 83.93 |
| LoRA (Hu et al., 2022) | 67.35 | 72.77 | 78.64 | 80.77 | 58.37 | 64.61 | 67.04 | 69.26 | 82.10 | 82.92 | 83.45 | 84.36 |
| Adapter (Houlsby et al., 2019) | 68.87 | 74.92 | 81.81 | 82.80 | 62.30 | 66.36 | 68.52 | 70.29 | 84.75 | 84.97 | 85.37 | 86.55 |
| Adapterformer (Chen et al., 2022) | 68.53 | 74.49 | 81.73 | 82.61 | 62.28 | 66.00 | 68.76 | 70.50 | 84.69 | 84.92 | 85.16 | 86.15 |
| Convpass (Jie et al., 2024) | 70.20 | 75.84 | 82.13 | 83.22 | 66.04 | 68.22 | 69.80 | 71.19 | 85.26 | 85.70 | 85.80 | 86.97 |
| CIAT (Zhu et al., 2021) | 68.17 | 73.60 | 80.66 | 81.27 | 57.89 | 64.61 | 68.17 | 70.01 | 82.92 | 83.33 | 83.95 | 85.43 |
| AIM (Yang et al., 2023) | 67.63 | 73.76 | 81.03 | 81.90 | 60.56 | 65.00 | 67.73 | 70.10 | 84.20 | 84.63 | 84.76 | 85.91 |
| **DKA** | **72.24** | **77.24** | **83.69** | **84.34** | **68.68** | **70.77** | **72.15** | **73.48** | **85.91** | **86.25** | **86.38** | **87.68** |

# D ADDITIONAL CLASSIFICATION RESULTS ON NATURAL-PRETRAINED MODELS

## D.1 CLASSIFICATION RESULTS WITH ADDITIONAL METRICS

In addition to the accuracy results reported in the Section 4.1, we further provide a comprehensive evaluation of `DKA` and baselines using F1 and sensitivity (SEN) scores on three classification datasets: COVID, BUSI, and ISIC-2019. As summarized in Table 8, `DKA` consistently achieves the best or near-best performance across all data scales (0.63%, 1.25%, and 100%). In particular, under low-data regimes, `DKA` exhibits notable improvements over existing PEFT approaches such as BitFit, LoRA, and Adapter-based methods, highlighting its robustness in capturing discriminative features when supervision is scarce. These results further validate the effectiveness of `DKA` beyond standard accuracy and confirm its ability to improve both predictive balance (F1) and clinical reliability (SEN) in medical image classification.

Table 8: **Comparison of Baselines and `DKA` on Three Classification Datasets Across Varying Data Sizes.** Experiments are based on the pretrained ViT-B. SEN is reported as percentages, while F1 is presented as raw value. The best results are highlighted in bold.

| Methods | COVID | | | BUSI | | | ISIC-2019 | | |
|---|---|---|---|---|---|---|---|---|---|
| | 0.63% | 1.25% | 100% | 0.63% | 1.25% | 100% | 0.63% | 1.25% | 100% |
| **F1** | | | | | | | | | |
| Full Fine-tuning | 0.845 | 0.855 | 0.970 | 0.697 | 0.752 | 0.939 | 0.293 | 0.317 | 0.539 |
| Linear Probing | 0.831 | 0.843 | 0.932 | 0.710 | 0.756 | 0.897 | 0.287 | 0.307 | 0.442 |
| BitFit (Zaken et al., 2021) | 0.718 | 0.762 | 0.950 | 0.542 | 0.573 | 0.879 | 0.184 | 0.235 | 0.501 |
| Prompt (Jia et al., 2022) | 0.753 | 0.819 | 0.972 | 0.588 | 0.618 | 0.930 | 0.218 | 0.235 | 0.534 |
| LoRA (Hu et al., 2022) | 0.786 | 0.814 | 0.973 | 0.603 | 0.641 | 0.945 | 0.214 | 0.235 | 0.528 |
| Adapter (Houlsby et al., 2019) | 0.802 | 0.833 | 0.978 | 0.607 | 0.704 | 0.931 | 0.231 | 0.248 | 0.501 |
| Adapterformer (Chen et al., 2022) | 0.793 | 0.807 | 0.974 | 0.614 | 0.693 | 0.918 | 0.228 | 0.231 | 0.495 |
| Convpass (Jie et al., 2024) | 0.807 | 0.833 | 0.969 | 0.622 | 0.718 | 0.938 | 0.253 | 0.275 | 0.512 |
| CIAT (Zhu et al., 2021) | 0.748 | 0.794 | 0.947 | 0.574 | 0.635 | 0.899 | 0.162 | 0.195 | 0.429 |
| AIM (Yang et al., 2023) | 0.776 | 0.804 | 0.953 | 0.602 | 0.682 | 0.907 | 0.182 | 0.227 | 0.433 |
| **DKA** | **0.865** | **0.881** | **0.981** | **0.720** | **0.774** | **0.951** | **0.296** | **0.326** | **0.542** |
| **SEN (%)** | | | | | | | | | |
| Full Fine-tuning | 83.57 | 84.15 | 97.24 | 69.27 | 74.26 | 93.46 | 27.76 | 30.64 | 52.35 |
| Linear Probing | 82.64 | 83.49 | 92.74 | 70.25 | 74.43 | 87.74 | 26.87 | 28.68 | 42.71 |
| BitFit (Zaken et al., 2021) | 70.36 | 75.34 | 94.45 | 51.33 | 54.41 | 85.52 | 18.50 | 21.74 | 47.55 |
| Prompt (Jia et al., 2022) | 74.31 | 81.76 | 96.57 | 56.71 | 59.35 | 91.34 | 20.65 | 23.92 | 49.28 |
| LoRA (Hu et al., 2022) | 77.42 | 80.76 | 96.76 | 59.37 | 61.41 | 92.07 | 20.11 | 24.13 | 50.75 |
| Adapter (Houlsby et al., 2019) | 81.38 | 82.40 | 97.41 | 59.40 | 66.69 | 92.58 | 21.84 | 24.93 | 49.50 |
| Adapterformer (Chen et al., 2022) | 79.34 | 81.51 | 97.40 | 60.96 | 65.44 | 90.72 | 20.36 | 22.78 | 48.31 |
| Convpass (Jie et al., 2024) | 81.23 | 82.71 | 95.76 | 61.79 | 67.90 | 92.74 | 24.63 | 27.36 | 51.12 |
| CIAT (Zhu et al., 2021) | 74.93 | 78.34 | 94.35 | 54.52 | 61.57 | 87.85 | 16.24 | 20.17 | 42.18 |
| AIM (Yang et al., 2023) | 76.04 | 80.76 | 94.50 | 58.75 | 65.46 | 89.45 | 18.06 | 21.07 | 47.16 |
| **DKA** | **84.41** | **87.76** | **98.12** | **71.82** | **76.78** | **95.46** | **28.20** | **31.50** | **53.69** |

## D.2 CLASSIFICATION RESULTS WITH MORE DATA SCALES

To further substantiate the advantages of DKA identified in the main text, we present additional classification results across a wider range of data scales (12.5% to 75%) in Table 9. These results reinforce the key findings, demonstrating that DKA consistently outperforms other baselines across all datasets (COVID, BUSI, ISIC-2019). This superior performance holds not only in extreme low-data regimes (0.63% and 1.25%) and full-data regimes (100%) but also across mid-scale settings, confirming that DKA maintains its advantage regardless of data availability. This performance reflects the effectiveness of its dual-branch design, which simultaneously captures local detail and broader context, providing a critical edge in medical image classification.

Table 9: **Additional Classification Results on Three Datasets with Varying Training Set Ratios.** Experiments are based on the pretrained ViT-B. ACC and SEN are reported as percentages, while F1 is presented as raw value. The best results are highlighted in bold.

| Methods | COVID | | | | BUSI | | | | ISIC-2019 | | | |
|---|---|---|---|---|---|---|---|---|---|---|---|---|
| | 12.5% | 25% | 50% | 75% | 12.5% | 25% | 50% | 75% | 12.5% | 25% | 50% | 75% |
| ACC (%) | | | | | | | | | | | | |
| Full Fine-tuning | 95.90 | 97.10 | 97.92 | 98.28 | 85.42 | 90.01 | 91.32 | 93.00 | 69.81 | 72.70 | 76.57 | 79.06 |
| Linear Probing | 92.84 | 93.43 | 94.16 | 94.34 | 79.23 | 84.03 | 86.54 | 87.22 | 63.66 | 65.75 | 66.60 | 68.28 |
| BitFit (Zaken et al., 2021) | 83.02 | 90.51 | 93.57 | 93.50 | 69.01 | 75.00 | 84.12 | 90.98 | 60.96 | 63.17 | 66.33 | 72.10 |
| Prompt (Jia et al., 2022) | 87.32 | 94.06 | 96.38 | 97.53 | 72.52 | 80.40 | 87.91 | 91.15 | 62.46 | 67.24 | 69.67 | 75.09 |
| LoRA (Hu et al., 2022) | 88.50 | 95.88 | 97.46 | 98.43 | 77.45 | 85.50 | 90.77 | 93.25 | 62.62 | 68.15 | 73.47 | 77.56 |
| Adapter (Houlsby et al., 2019) | 94.32 | 96.04 | 97.14 | 97.67 | 80.51 | 86.18 | 89.78 | 90.73 | 65.54 | 68.51 | 70.55 | 73.41 |
| Adapterformer (Chen et al., 2022) | 90.00 | 92.19 | 94.67 | 97.39 | 78.00 | 85.19 | 87.75 | 89.37 | 63.94 | 69.45 | 69.73 | 71.88 |
| Convpass (Jie et al., 2024) | 94.89 | 96.48 | 97.32 | 97.73 | 81.79 | 86.89 | 89.65 | 90.94 | 66.96 | 69.56 | 71.27 | 74.25 |
| CIAT (Zhu et al., 2021) | 88.98 | 92.73 | 94.23 | 95.45 | 72.47 | 79.32 | 83.56 | 87.55 | 59.84 | 65.16 | 67.55 | 70.25 |
| AIM (Yang et al., 2023) | 89.38 | 93.88 | 95.76 | 96.17 | 76.61 | 82.00 | 85.63 | 86.92 | 65.62 | 69.24 | 70.82 | 73.58 |
| **DKA** | **96.86** | **97.34** | **98.29** | **98.55** | **87.26** | **91.10** | **93.73** | **95.01** | **70.47** | **74.04** | **77.42** | **80.06** |
| F1 | | | | | | | | | | | | |
| Full Fine-tuning | 0.930 | 0.951 | 0.956 | 0.968 | 0.833 | 0.874 | 0.902 | 0.925 | 0.402 | 0.435 | 0.481 | 0.502 |
| Linear Probing | 0.898 | 0.916 | 0.922 | 0.929 | 0.782 | 0.828 | 0.856 | 0.861 | 0.335 | 0.351 | 0.379 | 0.392 |
| BitFit (Zaken et al., 2021) | 0.812 | 0.860 | 0.903 | 0.917 | 0.636 | 0.743 | 0.821 | 0.875 | 0.276 | 0.344 | 0.363 | 0.411 |
| Prompt (Jia et al., 2022) | 0.848 | 0.921 | 0.938 | 0.953 | 0.683 | 0.795 | 0.862 | 0.905 | 0.319 | 0.384 | 0.397 | 0.446 |
| LoRA (Hu et al., 2022) | 0.866 | 0.934 | 0.956 | 0.965 | 0.747 | 0.847 | 0.886 | 0.932 | 0.331 | 0.387 | 0.439 | 0.462 |
| Adapter (Houlsby et al., 2019) | 0.920 | 0.946 | 0.958 | 0.967 | 0.783 | 0.842 | 0.871 | 0.897 | 0.354 | 0.382 | 0.423 | 0.457 |
| Adapterformer (Chen et al., 2022) | 0.883 | 0.905 | 0.920 | 0.953 | 0.767 | 0.826 | 0.856 | 0.873 | 0.340 | 0.408 | 0.416 | 0.427 |
| Convpass (Jie et al., 2024) | 0.923 | 0.941 | 0.958 | 0.962 | 0.794 | 0.840 | 0.878 | 0.907 | 0.372 | 0.409 | 0.421 | 0.458 |
| CIAT (Zhu et al., 2021) | 0.843 | 0.905 | 0.911 | 0.925 | 0.708 | 0.788 | 0.813 | 0.844 | 0.293 | 0.364 | 0.375 | 0.417 |
| AIM (Yang et al., 2023) | 0.869 | 0.917 | 0.928 | 0.944 | 0.731 | 0.813 | 0.839 | 0.842 | 0.334 | 0.404 | 0.415 | 0.425 |
| **DKA** | **0.949** | **0.958** | **0.970** | **0.978** | **0.856** | **0.891** | **0.931** | **0.943** | **0.407** | **0.456** | **0.488** | **0.514** |
| SEN (%) | | | | | | | | | | | | |
| Full Fine-tuning | 93.24 | 95.16 | 95.52 | 96.68 | 82.24 | 87.69 | 89.92 | 91.05 | 38.84 | 40.01 | 41.56 | 49.36 |
| Linear Probing | 88.72 | 90.13 | 90.85 | 92.06 | 76.50 | 81.25 | 84.31 | 85.28 | 32.67 | 34.81 | 36.12 | 38.21 |
| BitFit (Zaken et al., 2021) | 79.33 | 85.27 | 89.38 | 91.15 | 63.85 | 72.62 | 80.77 | 86.31 | 28.23 | 35.85 | 41.62 | 44.38 |
| Prompt (Jia et al., 2022) | 83.56 | 91.52 | 93.24 | 94.95 | 68.27 | 77.25 | 85.47 | 90.54 | 31.54 | 37.63 | 43.10 | 46.75 |
| LoRA (Hu et al., 2022) | 85.15 | 92.90 | 95.65 | 96.22 | 71.12 | 81.34 | 87.93 | 90.23 | 31.88 | 37.31 | 42.61 | 46.90 |
| Adapter (Houlsby et al., 2019) | 91.93 | 93.65 | 95.36 | 96.29 | 76.53 | 85.78 | 87.39 | 88.03 | 34.65 | 37.31 | 41.16 | 44.12 |
| Adapterformer (Chen et al., 2022) | 87.35 | 89.80 | 91.04 | 95.54 | 73.68 | 82.74 | 85.79 | 87.16 | 32.52 | 39.10 | 39.75 | 41.56 |
| Convpass (Jie et al., 2024) | 92.00 | 94.51 | 94.92 | 95.36 | 76.78 | 80.73 | 85.03 | 88.21 | 35.36 | 39.84 | 41.58 | 43.99 |
| CIAT (Zhu et al., 2021) | 84.52 | 89.49 | 91.72 | 93.61 | 69.64 | 76.49 | 80.83 | 84.21 | 21.79 | 24.31 | 36.25 | 36.50 |
| AIM (Yang et al., 2023) | 86.51 | 90.22 | 92.53 | 93.27 | 73.86 | 79.80 | 83.77 | 84.27 | 34.23 | 37.80 | 40.65 | 42.62 |
| **DKA** | **95.24** | **96.27** | **97.18** | **97.59** | **83.88** | **89.62** | **93.46** | **93.89** | **39.72** | **44.07** | **47.50** | **50.39** |

# E  ADDITIONAL RESULTS ON MEDICAL-PRETRAINED MODELS

To provide more comprehensive evidence beyond the main paper, we further report detailed results using medical-pretrained models. Table 10 presents classification performance on the ISIC-2019 dataset with RadImageNet-pretrained ResNet-50 (Mei et al., 2022), where DKA consistently improves over linear probing and standard adapter tuning across different training ratios in terms of ACC, F1, and SEN. Table 11 reports segmentation results on the BUSI dataset with MedSAM (Ma et al., 2024), showing that DKA achieves clear gains over both baselines in terms of mIoU and Dice. These extended results corroborate our main findings in Section 4.5, demonstrating that the advantages of DKA generalize robustly to medical-pretrained models and are not restricted to natural-pretrained backbones.

Table 10: **Additional Classification Results based on the RadImageNet-pretrained ResNet-50 backbone on the ISIC-2019 dataset under varying training ratios.** SEN is reported as percentages, while F1 is presented as raw value. The best results are highlighted in bold.

| Methods | 0.63% | 1.25% | 100% |
|---|---|---|---|
| **F1** | | | |
| Full Fine-tuning | 0.223 | 0.262 | 0.483 |
| Linear Probing | 0.216 | 0.236 | 0.387 |
| BitFit | 0.172 | 0.208 | 0.411 |
| Prompt | 0.200 | 0.232 | 0.439 |
| LoRA | 0.203 | 0.230 | 0.431 |
| Adapter | 0.218 | 0.242 | 0.440 |
| DKA | **0.230** | **0.278** | **0.498** |
| **SEN (%)** | | | |
| Full Fine-tuning | 22.57 | 26.86 | 43.14 |
| Linear Probing | 20.28 | 24.52 | 37.61 |
| BitFit | 17.69 | 22.86 | 40.20 |
| Prompt | 19.83 | 24.42 | 44.05 |
| LoRA | 19.79 | 25.77 | 44.38 |
| Adapter | 20.34 | 25.13 | 44.57 |
| DKA | **23.40** | **27.89** | **45.39** |

Table 11: **Additional Segmentation Results based on MedSAM backbone on the BUSI segmentation task under varying training ratios.** Results are reported in terms of Dice (%). The best results are highlighted in bold.

| Methods | 0.63% | 1.25% | 100% |
|---|---|---|---|
| **Dice (%)** | | | |
| Full Fine-tuning | 54.54 | 63.08 | 81.15 |
| Linear Probing | 50.09 | 60.90 | 79.68 |
| BitFit | 23.46 | 47.31 | 74.15 |
| Prompt | 25.19 | 50.65 | 78.82 |
| LoRA | 32.58 | 55.03 | 80.42 |
| Adapter | 52.28 | 61.81 | 80.35 |
| DKA | **55.39** | **65.24** | **83.97** |

## F  ADDITIONAL ABLATIONS

### F.1  DKA IN DIFFERENT BACKBONE

To evaluate whether the advantages of DKA transfer to other architectures, we repeat classification experiments on the pretrained Swin-B using the same datasets: COVID, BUSI, and ISIC-2019. As shown in Figure 9, DKA consistently outperforms full fine-tuning and all PEFT baselines across different data scales. Critically, while other PEFT methods struggle to match linear probing, and often degrade substantially in constrained-data regimes, DKA maintains strong performance in both settings.

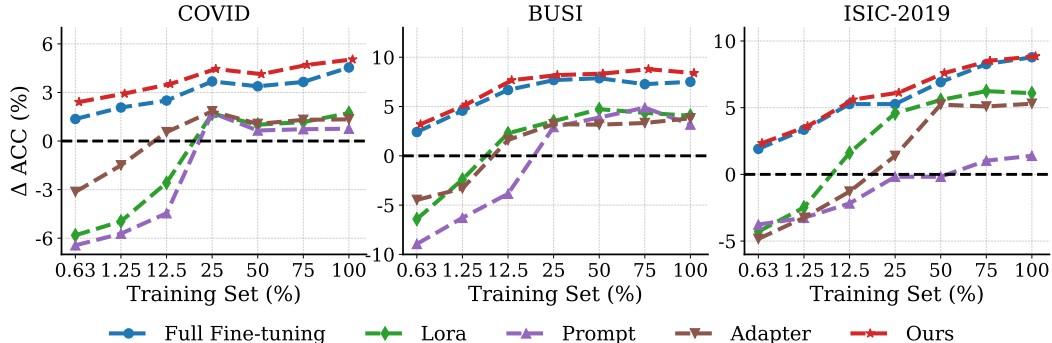

Figure 9: **Performance of Baselines and DKA across Various Training Data Sizes.** $\Delta$ACC = $\text{ACC}_{Baslines} - \text{ACC}_{LinearProbing}$. Experiments are based on the pretrained Swin-B for COVID, BUSI, and ISIC-2019.

### F.2  PERFORMANCE OF DUAL-KERNEL CONVOLUTION AND ASYNCHRONOUS LEARNING RATES

To further evaluate the impact of our proposed enhancements, including the introduction of dual-convolution design and a learning rate split strategy within the DKA module, we present additional results in Table 12. This table compares DKA against two baselines (Adapter + Dual-Conv and Adapter + LR Split) across multiple datasets (COVID, BUSI, ISIC-2019) and training set ratios. Consistently, DKA outperforms both baselines, achieving the highest overall accuracy in each dataset, with notable gains observed even in low-data regimes. This confirms the effectiveness of our combined strategy in capturing richer local-global features and improving learning stability.

Table 12: **Performance Comparison of Enhanced Adapter Designs.** Experiments are conducted on the pretrained ViT-B across three medical imaging classification datasets and varying training set ratios by reporting ACC (%).

| Datasets | Methods | Training Set Size | | | | | | |
|---|---|---|---|---|---|---|---|---|
| | | 0.63% | 1.25% | 12.5% | 25% | 50% | 75% | 100% |
| COVID | Adapter + Dual-Conv | 86.78 | 88.89 | 96.01 | 96.98 | 98.13 | 97.95 | 98.61 |
| | Adapter + LR Split | 88.04 | 88.95 | 96.20 | 97.04 | 97.64 | 98.07 | 98.79 |
| | Ours | **89.01** | **91.06** | **96.86** | **97.34** | **98.29** | **98.55** | **99.21** |
| BUSI | Adapter + Dual-Conv | 67.73 | 77.32 | 83.39 | 89.14 | 90.73 | 91.05 | 94.25 |
| | Adapter + LR Split | 70.37 | 77.64 | 84.71 | 89.46 | 91.10 | 93.37 | 94.93 |
| | Ours | **74.23** | **79.64** | **87.26** | **91.10** | **93.73** | **95.01** | **95.89** |
| ISIC-2019 | Adapter + Dual-Conv | 59.49 | 60.51 | 70.12 | 73.27 | 76.60 | 78.52 | 82.25 |
| | Adapter + LR Split | 59.06 | 60.09 | 69.99 | 72.24 | 76.07 | 78.52 | 81.67 |
| | Ours | **60.52** | **62.32** | **70.47** | **74.04** | **77.42** | **80.06** | **83.09** |

## F.3 DKA POSITION

To understand how the position of inserted DKA influences performance, we explore three placement strategies using the pretrained ViT-B with 12 transformer blocks: inserting DKA into the bottom 4 blocks (Blocks 0–3), middle 4 blocks (Blocks 4–7), and top 4 blocks (Blocks 8–11). We also include a reference setting where DKA is inserted into all layers. We extend our analysis to three medical imaging classification datasets: COVID, BUSI, and ISIC-2019. As shown in Table 13, the position of DKA significantly affects model performance across various training set sizes. Among partial configurations, placing DKA in the middle layers consistently outperforms the top and bottom placements across datasets. Notably, under low-data regimes (e.g., 0.63%), placing DKA in the top layers offers stronger performance than bottom or middle placement, highlighting the value of adapting higher-level representations when supervision is scarce. This trend is consistent with observations from prior work (Yang et al., 2023), which reported that inserting adapters in bottom blocks yields limited performance.

Table 13: **Effect of Position.** Experiments are conducted on the pretrained ViT-B across three medical imaging classification datasets and varying training set ratios by reporting ACC (%) with standard deviations.

| Datasets | Positions | Training Set Size | | | | | | |
|---|---|---|---|---|---|---|---|---|
| | | 0.63% | 1.25% | 12.5% | 25% | 50% | 75% | 100% |
| COVID | Bottom | 86.04 ± 1.61 | 88.68 ± 1.38 | 93.69 ± 0.89 | 94.29 ± 0.75 | 95.52 ± 0.60 | 96.64 ± 0.45 | 97.34 ± 0.35 |
| | Middle | 86.80 ± 1.45 | 89.18 ± 1.25 | 95.43 ± 0.53 | 96.80 ± 0.41 | 97.76 ± 0.29 | 98.43 ± 0.25 | 98.79 ± 0.21 |
| | Top | 87.78 ± 1.21 | 90.50 ± 1.13 | 94.72 ± 0.69 | 96.12 ± 055 | 97.22 ± 0.38 | 98.23 ± 0.32 | 98.53 ± 0.26 |
| | ALL | **89.01 ± 0.90** | **91.06 ± 0.84** | **96.86 ± 0.40** | **97.34 ± 0.32** | **98.29 ± 0.26** | **98.55 ± 0.24** | **99.21 ± 0.13** |
| BUSI | Bottom | 71.47 ± 2.06 | 77.16 ± 1.47 | 84.86 ± 1.11 | 88.00 ± 0.65 | 91.19 ± 0.43 | 92.79 ± 0.33 | 93.94 ± 0.27 |
| | Middle | 73.48 ± 1.79 | 78.22 ± 1.28 | 86.62 ± 0.75 | 90.43 ± 0.44 | 93.43 ± 0.25 | 94.94 ± 0.19 | 95.46 ± 0.11 |
| | Top | 73.86 ± 1.64 | 78.16 ± 1.22 | 86.48 ± 0.84 | 90.08 ± 0.45 | 93.07 ± 0.28 | 94.62 ± 0.20 | 95.00 ± 0.13 |
| | ALL | **74.23 ± 1.53** | **79.64 ± 1.17** | **87.26 ± 0.64** | **91.10 ± 0.39** | **93.73 ± 0.23** | **95.01 ± 0.12** | **95.89 ± 0.09** |
| ISIC-2019 | Bottom | 56.87 ± 2.47 | 58.68 ± 2.11 | 67.28 ± 1.52 | 70.37 ± 1.26 | 73.43 ± 1.05 | 76.16 ± 0.73 | 79.52 ± 0.46 |
| | Middle | 57.64 ± 2.33 | 60.01 ± 2.05 | 69.88 ± 1.34 | 73.95 ± 1.06 | 76.74 ± 0.74 | 79.43 ± 0.43 | 82.59 ± 0.20 |
| | Top | 59.89 ± 2.17 | 61.79 ± 1.95 | 69.32 ± 1.38 | 72.59 ± 1.18 | 76.04 ± 0.79 | 78.42 ± 0.52 | 81.90 ± 0.25 |
| | ALL | **60.52 ± 2.02** | **62.32 ± 1.85** | **70.47 ± 1.26** | **74.04 ± 0.99** | **77.42 ± 0.67** | **80.06 ± 0.36** | **83.09 ± 0.14** |

## F.4 ERF OF ADAPTER IN DIFFERENT BACKBONE

To further validate our findings on the impact of data scarcity on Adapter performance, we extend the effective receptive field (ERF) analysis to the pretrained Swin-T model (Liu et al., 2021), as shown in Figure 10. This complementary analysis reinforces our observations from the pretrained ViT-B model (Figure 2), revealing a similar contraction of ERFs as the training set size decreases. Specifically, under extreme data scarcity (e.g., 0.63% and 1.25% training data), the pretrained Swin-T exhibits a sharply reduced ERF, aligning with the earlier findings (see Section 2) that Adapters can disrupt pretrained feature representations under severe data constraints. This result suggests that the negative impacts observed in the main text are not limited to a single architecture but are likely a more general phenomenon affecting a wide range of vision backbones.

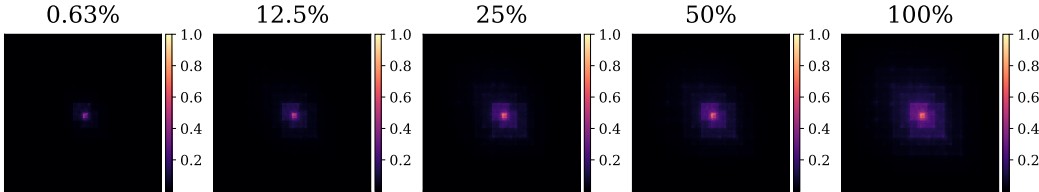

Figure 10: **Effective Receptive Field of Adapter under Different Training Set Ratios.** Experiments are conducted on the COVID dataset using the pretrained Swin-T.

## F.5 Effect of Dilated vs. Standard Kernels

We further compare dilated convolutional kernels with our standard large-kernel design. In particular, we evaluate several dilated kernel settings, including $3 \times 3$ with dilation rates of $d = 3, 25$, $11 \times 11$ with $d = 5$, and $26 \times 26$ with $d = 2$, against the dual-kernel configuration of $5 \times 5 + 51 \times 51$. The classification results on the BUSI dataset are shown in Table 14, and the segmentation results on the ISIC-2018 dataset are reported in Table 15.

Across both datasets, the $5 \times 5 + 51 \times 51$ design consistently achieves the best performance in terms of ACC, F1, and SEN for classification, as well as mIoU and Dice for segmentation. Although dilated kernels provide a larger effective receptive field, they underperform compared to explicitly using a large standard kernel. This suggests that our dual-kernel design captures both local and global information more effectively than dilated alternatives, highlighting the efficiency of standard large kernels in the `DKA` framework.

Table 14: **Comparison of different kernel designs on the BUSI dataset for classification under varying training ratios.** Experiments are based on the pretrained ViT-B. ACC and SEN are reported as percentages, while F1 is presented as raw value. The best results are highlighted in bold. "$d$" represents the dilation rate in dilated convolution.

| Kernel Designs | 0.63% | 1.25% | 100% |
|---|---|---|---|
| ACC (%) | | | |
| 3×3 ($d$=3) + 3×3 ($d$=25) | 66.77 | 77.32 | 93.61 |
| 3×3 ($d$=3) + 11×11 ($d$=5) | 67.41 | 77.64 | 94.25 |
| 3×3 ($d$=3) + 26×26 ($d$=2) | 68.69 | 77.96 | 94.37 |
| 5×5 + 51×51 | **74.23** | **79.46** | **95.89** |
| F1 | | | |
| 3×3 ($d$=3) + 3×3 ($d$=25) | 0.641 | 0.751 | 0.931 |
| 3×3 ($d$=3) + 11×11 ($d$=5) | 0.650 | 0.756 | 0.936 |
| 3×3 ($d$=3) + 26×26 ($d$=2) | 0.663 | 0.758 | 0.942 |
| 5×5 + 51×51 | **0.720** | **0.774** | **0.951** |
| SEN (%) | | | |
| 3×3 ($d$=3) + 3×3 ($d$=25) | 62.81 | 74.35 | 92.36 |
| 3×3 ($d$=3) + 11×11 ($d$=5) | 63.99 | 74.43 | 93.44 |
| 3×3 ($d$=3) + 26×26 ($d$=2) | 64.72 | 74.71 | 94.03 |
| 5×5 + 51×51 | **71.82** | **76.78** | **95.46** |

Table 15: **Comparison of different kernel designs on the ISIC-2018 dataset for segmentation under varying training ratios.** Experiments are based on the pretrained Segmenter-B. mIoU and Dice are reported as percentages. The best results are highlighted in bold. "$d$" represents the dilation rate in dilated convolution.

| Kernel Designs | 0.63% | 1.25% | 100% |
|---|---|---|---|
| mIoU (%) | | | |
| 3×3 ($d$=3) + 3×3 ($d$=25) | 60.25 | 72.71 | 76.43 |
| 3×3 ($d$=3) + 11×11 ($d$=5) | 61.46 | 73.11 | 76.97 |
| 3×3 ($d$=3) + 26×26 ($d$=2) | 62.27 | 73.43 | 77.48 |
| 5×5 + 51×51 | **63.13** | **74.27** | **78.53** |
| Dice (%) | | | |
| 3×3 ($d$=3) + 3×3 ($d$=25) | 75.19 | 84.20 | 86.64 |
| 3×3 ($d$=3) + 11×11 ($d$=5) | 76.13 | 84.46 | 86.99 |
| 3×3 ($d$=3) + 26×26 ($d$=2) | 76.75 | 84.68 | 87.31 |
| 5×5 + 51×51 | **77.39** | **85.24** | **87.97** |

F.6    Effect of Diverse Kernel Combinations

To further analyze the contribution of different kernel configurations in DKA, we evaluate multiple kernel combinations for both classification and segmentation tasks. Specifically, we compare three settings: (*i*) $5 \times 5 + 11 \times 11 + 51 \times 51$, (*ii*) $5 \times 5 + 31 \times 31 + 51 \times 51$, and (*iii*) $5 \times 5 + 51 \times 51$. Table 16 reports results on the BUSI classification dataset, while Table 17 presents results on the ISIC-2018 segmentation dataset.

Across both tasks, we observe that the dual-kernel configuration ($5 \times 5 + 51 \times 51$) consistently achieves the best trade-off, outperforming the three-branch alternatives in terms of ACC, F1, and SEN for classification, as well as mIoU and Dice for segmentation. These results indicate that adding intermediate kernels (e.g., $11 \times 11$ or $31 \times 31$) does not provide additional benefits, and a simpler dual-kernel design is sufficient to capture both local and global dependencies. This further validates the efficiency of our proposed kernel selection strategy within DKA.

Table 16: **Comparison of diverse kernel combinations on the BUSI dataset for classification under varying training ratios.** Experiments are based on the pretrained ViT-B. ACC and SEN are reported as percentages, while F1 is presented as raw value. The best results are highlighted in bold.

| Kernel Combinations | 0.63% | 1.25% | 100% |
|---|---|---|---|
| ACC (%) | | | |
| 5×5 +11×11 + 51×51 | 72.68 | 77.00 | 94.89 |
| 5×5 +31×31 + 51×51 | 71.04 | 76.68 | 94.57 |
| 5×5 + 51×51 | **74.23** | **79.46** | **95.89** |
| F1 | | | |
| 5×5 +11×11 + 51×51 | 0.709 | 0.743 | 0.942 |
| 5×5 +31×31 + 51×51 | 0.694 | 0.738 | 0.936 |
| 5×5 + 51×51 | **0.720** | **0.774** | **0.951** |
| SEN (%) | | | |
| 5×5 +11×11 + 51×51 | 69.17 | 73.56 | 94.11 |
| 5×5 +31×31 + 51×51 | 68.39 | 72.85 | 93.62 |
| 5×5 + 51×51 | **71.82** | **76.78** | **95.46** |

Table 17: **Comparison of diverse kernel combinations on the ISIC-2018 dataset for segmentation under varying training ratios.** Experiments are based on the pretrained Segmenter-B. mIoU and Dice are reported as percentages. The best results are highlighted in bold.

| Kernel Combinations | 0.63% | 1.25% | 100% |
|---|---|---|---|
| mIoU (%) | | | |
| 5×5 +11×11 + 51×51 | 62.72 | 73.76 | 78.06 |
| 5×5 +31×31 + 51×51 | 62.40 | 73.63 | 77.54 |
| 5×5 + 51×51 | **63.13** | **74.27** | **78.53** |
| Dice (%) | | | |
| 5×5 +11×11 + 51×51 | 77.09 | 84.90 | 87.68 |
| 5×5 +31×31 + 51×51 | 76.84 | 84.81 | 87.35 |
| 5×5 + 51×51 | **77.39** | **85.24** | **87.97** |

### F.7 EFFECT UNDER EXTREME LOW-DATA REGIMES

To further investigate the limits of DKA, we conduct experiments under an extreme low-data regime with only 0.125% of the training set available. This setup approximately corresponds to a 5-shot setting for COVID classification and genuine 1-shot settings for BUSI and ISIC-2019 (classification), as well as BRATS, BUSI, and ISIC-2018 (segmentation). The results are reported in Table 18 and Table 19.

Across all datasets, DKA consistently outperforms both Linear Probing and standard Adapter, even under such highly constrained supervision. For example, on ISIC-2019 classification, DKA improves ACC from 52.54% (Linear Probing) to 55.95%, and F1 from 0.227 to 0.263. Similarly, on BUSI segmentation, DKA achieves a Dice of 37.43%, substantially higher than Linear Probing (23.02%) and Adapter (24.93%). These results confirm that the proposed large-kernel design remains robust and effective even in the most challenging few-shot scenarios, highlighting its practicality for data-scarce medical applications.

Table 18: **Comparison of Linear Probing, Adapter, and DKA on three classification datasets under 0.125% training data.** Experiments are based on the pretrained ViT-B. ACC and SEN are reported as percentages, while F1 is presented as raw value. The best results are highlighted in bold.

| Methods | COVID | BUSI | ISIC-2019 |
|---|---|---|---|
| **ACC (%)** | | | |
| Linear Probing | 76.51 | 37.06 | 52.54 |
| Adapter | 71.38 | 35.46 | 47.25 |
| DKA | **78.74** | **39.62** | **55.95** |
| **F1** | | | |
| Linear Probing | 0.735 | 0.273 | 0.227 |
| Adapter | 0.704 | 0.247 | 0.153 |
| DKA | **0.787** | **0.340** | **0.263** |
| **SEN (%)** | | | |
| Linear Probing | 72.53 | 33.87 | 21.58 |
| Adapter | 68.40 | 30.41 | 15.07 |
| DKA | **75.64** | **37.13** | **24.39** |

Table 19: **Comparison of Linear Probing, Adapter, and DKA on three segmentation datasets under 0.125% training data.** Experiments are based on the pretrained Segmenter-B. mIoU and Dice are reported as percentages. The best results are highlighted in bold.

| Methods | BRATS | BUSI | ISIC-2018 |
|---|---|---|---|
| **mIoU (%)** | | | |
| Linear Probing | 3.25 | 19.64 | 53.21 |
| Adapter | 1.20 | 14.24 | 47.05 |
| DKA | **6.36** | **32.83** | **60.25** |
| **Dice (%)** | | | |
| Linear Probing | 6.94 | 23.02 | 69.46 |
| Adapter | 2.40 | 24.93 | 63.99 |
| DKA | **11.96** | **37.43** | **75.19** |

## F.8 EFFECT ON MEDICAL-PRETRAINED VISION-LANGUAGE MODELS.

To further validate the generalization ability of DKA, we extend our evaluation to medical vision-language models. Specifically, we adopt MedCLIP (Wang et al., 2022) as the backbone and follow the few-shot image classification protocol commonly used in prior works on vision-language adaptation (Shakeri et al., 2024; Silva-Rodriguez et al., 2024). We report results on the BUSI dataset under 1-shot, 4-shot, and 8-shot settings, where each configuration is averaged over five random seeds. As summarized in Table 20, DKA consistently surpasses linear probing and standard adapter tuning across all support sizes. These results demonstrate that the benefits of DKA are not limited to vision-only large pretrained models, but also extend to multimodal vision-language models, highlighting its robustness in broader medical AI scenarios.

Table 20: **Comparison of different methods under few-shot settings on the BUSI dataset based on MedCLIP.** Results are reported in terms of ACC, F1, and SEN.

| Tuning Strategies | 1-shot | 4-shot | 8-shot |
|---|---|---|---|
| ACC (%) | | | |
| Linear Probing | 66.25 | 74.21 | 78.41 |
| Adapter | 65.86 | 74.08 | 79.32 |
| DKA | **71.20** | **79.82** | **82.27** |
| F1 | | | |
| Linear Probing | 0.631 | 0.712 | 0.776 |
| Adapter | 0.625 | 0.706 | 0.782 |
| DKA | **0.697** | **0.762** | **0.815** |
| SEN (%) | | | |
| Linear Probing | 62.55 | 69.72 | 77.45 |
| Adapter | 61.87 | 69.46 | 78.96 |
| DKA | **68.17** | **75.85** | **81.87** |

## F.9 FREQUENCY-DOMAIN ANALYSIS OF KERNEL SIZES

To better understand the complementary roles of small and large kernels in DKA, we analyze their frequency responses using the radial power spectral density (PSD). Specifically, we compute the spectral centroid $f_c$ and the normalized frequency $f_{90}$ that captures 90% of the cumulative energy. Table 21 shows the results for $5 \times 5$ and $51 \times 51$ kernels. The smaller kernel exhibits a higher spectral centroid ($f_c = 0.618$ vs. $0.503$), indicating stronger sensitivity to high-frequency details. In contrast, the larger kernel shifts energy towards lower frequencies, facilitating global context modeling. This analysis provides further evidence for the effectiveness of combining diverse kernel sizes in DKA.

Table 21: Frequency-domain analysis of kernel sizes. Results are reported as mean±std over different trained models.

| Kernel Size | $f_c$ | $f_{90}$ |
|---|---|---|
| $5 \times 5$ | $0.618 \pm 0.018$ | $0.924 \pm 0.011$ |
| $51 \times 51$ | $0.503 \pm 0.004$ | $0.904 \pm 0.003$ |

## F.10 INFERENCE LATENCY AND MEMORY USAGE

We also report the inference efficiency of different methods in terms of latency and memory consumption on the BUSI dataset using the pretrained ViT-B. As summarized in Table 22, DKA introduces only marginal overhead compared to the standard adapter framework, with inference latency increasing by less than 0.5 ms and memory usage by less than 6 MB. These differences are negligible in practice, indicating that the proposed large-kernel design achieves substantial performance gains with minimal computational cost.

Table 23: **Comparison of methods on the BUSI segmentation dataset with batch size $= 4$ based on the pretrained Segmenter-B.**

| Methods | 0.63% | 1.25% | 100% |
|---|---|---|---|
| Full Fine-tuning | 27.02 | 33.11 | 57.91 |
| Linear Probing | 25.81 | 32.47 | 54.60 |
| Adapter | 18.67 | 26.50 | 55.46 |
| DKA | **27.26** | **35.21** | **59.35** |

Table 22: **Comparison of inference latency and memory usage on the BUSI dataset based on the pretrained ViT-B.** DKA introduces only negligible overhead compared to standard adapter variants.

| Methods | Inference Latency (ms) | Memory (MB) |
|---|---|---|
| Linear Probing | 6.78 | 352.81 |
| Adapter | 11.54 | 433.70 |
| Adapter + 5×5 Conv | 11.66 | 433.90 |
| Adapter + 51×51 Conv | 11.69 | 439.56 |
| DKA | **11.97** | **439.63** |

## F.11 EFFECTS OF LARGER BATCH SIZE

We further conducted additional segmentation experiments using a larger batch size of 4 under the same configuration. As shown in Table 23, increasing the batch size does not change the relative ranking: DKA consistently outperforms Full Fine-tuning, Linear Probing, and Adapter across all label ratios. This confirms that our improvements are robust to batch-size variations and are not tied to a specific optimization setting.

## F.12 PARAMETER EFFICIENCY ANALYSIS

The DKA module integrates two depthwise convolution branches with kernel sizes $k_1$ and $k_2$ within each adapter. These convolutions are applied independently on each of the $\hat{d}$ channels, contributing $\hat{d}(k_1^2 + k_2^2)$ parameters per module. Since DKA is inserted twice in every Transformer block (after the attention and feedforward layers), the total number of additional parameters grows linearly with the number of blocks. Even with two kernels (e.g., $k_1 = 51$, $k_2 = 5$), the total number of trainable parameters introduced by all DKA modules remains less than 2% of the pretrained backbone, maintaining strong parameter efficiency while providing strong performance gains, especially in low-data regimes.

## F.13 COMPLEMENTARY CAM VISUALIZATION

To complement the ERF analysis, we additionally compare Grad-CAM (Selvaraju et al., 2017) responses of the standard Adapter and our DKA on BUSI. As shown in Figure 11, the standard Adapter exhibits diffuse and background-driven activations, whereas DKA produces coherent, lesion-centered responses. This confirms that the enlarged ERF of DKA captures meaningful contextual information rather than merely expanding activation range.

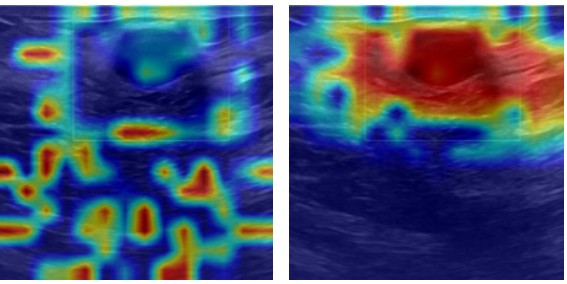

Figure 11: **Grad-CAM comparison on BUSI.** Left: standard Adapter; right: DKA with more focused, lesion-related activations.

# G PSEUDOCODE

**Algorithm 1** Pseudo-code of a Transformer block with DKA

```python
class DKA:
    def __init__(self, dim, middle_dim, kernel_large, kernel_small):
        self.downsample = Linear(dim, middle_dim)
        self.conv_large = DepthwiseConv(middle_dim, kernel_large)
        self.conv_small = DepthwiseConv(middle_dim, kernel_small)
        self.activation = GELU()
        self.upsample = Linear(middle_dim, dim)

    def forward(self, x):
        # Store the input for the residual connection
        residual = x
        x = self.downsample(x)

        # Dual-Path Convolutions (Large + Small)
        x_large = self.conv_large(x)
        x_small = self.conv_small(x)
        x = x_large + x_small

        x = self.activation(x)
        x = self.upsample(x)
        x = x + residual

        return x

class TransformerBlock_with_DKA:
    def __init__(self, dim, num_heads, mlp_ratio, middle_dim, kernel_large, kernel_small):
        # Original ViT components
        self.attn = MultiheadAttention(dim, num_heads)
        self.norm1 = LayerNorm(dim)
        self.mlp = MLP(dim, mlp_ratio)
        self.norm2 = LayerNorm(dim)

        # DKA Adapter
        self.dka = DKA(dim, middle_dim, kernel_large, kernel_small)

    def forward(self, x):
        residual = x
        x = self.norm1(x)
        x = self.attn(x)
        x = x + residual

        x = self.dka(x)

        residual = x
        x = self.norm2(x)
        x = self.mlp(x)
        x = x + residual

        x = self.dka(x)

        return x
```

