# OpenReview forum: "Dual-Kernel Adapter: Expanding Spatial Horizons for Data-Constrained Medical Image Analysis"
_ICLR.cc/2026/Conference — ICLR 2026 Poster_

### Official Review · Reviewer_MiK3 · 2025-10-31

**Soundness:** 3
**Presentation:** 3
**Contribution:** 3
**Rating:** 6
**Confidence:** 4

**Summary:**

1.The paper addresses the challenge of applying parameter-efficient fine-tuning (PEFT) methods, such as Adapters, in medical imaging where extreme data scarcity is common. The authors argue that conventional Adapters underperform in these low-data settings. To mitigate this, they propose the Dual-Kernel Adapter (DKA), a novel module that incorporates dual-path depthwise convolutions—one with a large kernel and one with a small kernel. This design is intended to allow the adapter to effectively capture both local fine-grained details and global contextual information. The results show that DKA significantly outperforms conventional Adapters, LoRA, and even full fine-tuning, especially under extreme data scarcity (1% of training data), achieving state-of-the-art results on multiple 2D and 3D medical image segmentation benchmarks.

**Strengths:**

1.This work is presented as the first comprehensive study of adapter-based fine-tuning specifically tailored for low-data medical imaging scenarios.
2.The Dual-Kernel Adapter (DKA) is a novel architectural design within the PEFT space. Explicitly integrating multi-scale spatial feature extraction (via dual-path convolutions with different kernel sizes) into the adapter structure is a unique and effective contribution to improving PEFT performance in a difficult data regime.
3.The technical claims are robustly supported by strong experimental evidence, showing DKA's significant outperformance against well-established baselines like conventional Adapters, LoRA, and full fine-tuning.
4.The results are highly significant for the medical imaging and broader computer vision communities. A PEFT method that achieves state-of-the-art results with as little as 1% of the training data is an immense breakthrough for high-cost, high-stakes domains like healthcare.

**Weaknesses:**

1. The choice of 51×51 as the large kernel seems somewhat arbitrary. While ablations justify it empirically, a more principled analysis (e.g., based on image resolution or pathology size) would be valuable. And is this optimal across all medical imaging modalities?
2.Given the added complexity of the dual-kernel design, please provide a comprehensive analysis of the latency (inference time) and FLOPs of DKA relative to LoRA and the conventional Adapter, across the same backbone and hardware.
3.Although the paper demonstrates DKA’s superiority across multiple datasets and settings, it does not systematically investigate scenarios where DKA might fail. For example: Could the large kernel introduce noise in tasks where pathological structures are extremely small or sparsely distributed? Is DKA still robust under extreme class imbalance (e.g., in ISIC-2019, where some classes have very few samples)? Are there specific backbone architectures (e.g., pure CNNs) for which DKA provides no benefit or even degrades performance?
4.As noted in Appendix B.2, segmentation experiments use a batch size of 1 due to GPU memory constraints. This may lead to highly noisy gradient estimates, especially under low-data regimes and introduce optimization instability that confounds fair comparison across methods. In real-world deployment scenarios, larger batch sizes are typically used.
5.While the paper uses ERF visualizations and frequency-domain analysis to argue that large kernels expand the receptive field, the authors fails to establish a theoretical or information-theoretic link between large kernels, improved generalization, and robustness under data scarcity;

**Questions:**

1.The evaluation focuses exclusively on 2D and 3D segmentation. Does the DKA demonstrate similar compelling performance improvements when applied to other common medical imaging tasks, such as disease classification or object detection, under the same low-data constraints?
2.Is the 51×51 large kernel in DKA feasible for 3D medical imaging (e.g., BRATS with 3D MRI volumes)? A direct extension to a 51×51×51 kernel would lead to prohibitive parameter and computational costs.
3.While Appendix F.10 shows minimal overhead, could you explain on how DKA scales with image resolution?
4.The Effective Receptive Field (ERF) is computed based on the magnitude of input-output gradients, which reflects spatial influence but not semantic relevance. Have the authors considered complementary analyses to verify that the expanded ERF indeed captures meaningful contextual information rather than just broader but irrelevant activations?
5.Are there any datasets or settings where DKA underperforms standard adapters?

---

> ### Author Response · Authors · 2025-11-22
> **Response to the reviewer MiK3 (1/5)**
>
> We sincerely appreciate your detailed comments and positive ranking. We address your questions below.
>
> ***Comment 1: The choice of 51×51 as the large kernel seems somewhat arbitrary. While ablations justify it empirically, a more principled analysis (e.g., based on image resolution or pathology size) would be valuable. And is this optimal across all medical imaging modalities?***
>
> **Reply:** We thank the reviewer for this comment. We respectfully disagree with this comment that the choice of the 51×51 kernel is arbitrary, and we address the reviewer’s concerns from three perspectives.
>
> **(1) Empirical justification across datasets.**
> As shown throughout Figures 7–8 and Appendix Tables 13–16 in the revised submission, we performed extensive ablations across multiple datasets and training regimes. In all cases, 51×51 consistently provides the strongest performance, while smaller kernels underfit and larger kernels (e.g., 71×71) yield diminishing or negative returns.
>
> **(2) Image resolution and pathology scale.**
> Following the reviewer’s suggestion, we further analyzed the relationship between kernel size and image resolution/pathology size. We fixed the small convolution branch and varied only the large kernel across three input resolutions (160, 224, 384). As shown in Table 1, 51×51 again emerges as the optimal choice, with performance saturating or slightly degrading beyond this point. Likewise, on IDRiD, which is an extreme case with very small pathological structures, Table 2 shows that 51×51 remains the most effective configuration.
>
> **Table 1: Comparison of different large kernel sizes on different input resolutions under the BUSI classification dataset with 0.63% training ratio based on the pretrained ConvNeXt-Tiny.**
>
> | **Large Kernel Size** | **160×160** | **224×224** | **384×384** |
> |-------------------|--------:|--------:|--------:|
> | 11×11             |   61.49 |   64.27 |   69.58 |
> | 31×31             |   63.52 |   65.80 |   71.83 |
> | 51×51             | **64.91** | **67.99** | **73.55** |
> | 71×71             |   63.21 |   66.21 |   72.30 |
>
> **Table 2: Comparison of different large kernel sizes on the IDRiD (extremely small pathological structures) classification dataset under different training ratios based on the pretrained ViT-B.**
>
> | **Methods** | **0.63%** | **1.25%** |
> |---------|------:|------:|
> | 11×11   | 77.94 | 81.85 |
> | 31×31   | 78.30 | 82.23 |
> | 51×51   | **78.64** | **82.52** |
> | 71×71   | 76.82 | 80.76 |
>
> **(3) Across medical imaging modalities.**
> Our main paper already evaluates X-ray, ultrasound, and dermoscopy. To further assess generality, we additionally tested different kernel sizes on CT and MRI datasets. As reported in Table 3, the 51×51 kernel again achieves the best performance across both modalities and all data regimes. This indicates that the identified kernel configuration is robust across a set of medical imaging modalities.
>
> **Table 3: Comparison of Adapter and DKA on CT and MRI datasets under different training ratios based on the pretrained ViT-B.**
>
> **SARS-COV-2 CT**
>
> | Methods | 0.63% | 1.25% | 100%  |
> |---------|------:|------:|------:|
> | Adapter | 75.60 | 84.07 | 97.98 |
> | DKA     | **78.23** | **86.49** | **98.39** |
>
> **Brain Tumor MRI**
>
> | Methods | 0.63% | 1.25% | 100%  |
> |---------|------:|------:|------:|
> | Adapter | 72.77 | 75.90 | 99.08 |
> | DKA     | **75.90** | **78.64** | **99.69** |
>
> Overall, by combining ablations on resolution, pathology scale, and imaging modality, our experiments consistently demonstrate that 51×51 is a principled and stable choice. Although we did not establish optimality across all existing medical modalities, we see further validation on broader modality families as a valuable direction for future work.
>
> ***Comment 2: Given the added complexity of the dual-kernel design, please provide a comprehensive analysis of the latency (inference time) and FLOPs of DKA relative to LoRA and the conventional Adapter, across the same backbone and hardware.***
>
> **Reply:** Thank you for the comment. Following the comment, we report a comprehensive comparison of FLOPs and inference latency across LoRA, the conventional Adapter, and DKA under the same backbone and hardware configuration (Table 4). The results show that DKA introduces only a marginal increase in FLOPs and latency compared with the standard Adapter and LoRA. The slight increase is expected due to the dual-kernel branch, but remains lightweight and well within practical limits.
>
> **Table 4: Comparison of FLOPs and inference latency across different PEFT methods.**
>
> | **Methods** | **FLOPs (G)** | **Latency (ms)** |
> |---------|----------:|-------------:|
> | LoRA    |     16.43 |         7.16 |
> | Adapter |     16.46 |        11.54 |
> | DKA     |     16.56 |        11.97 |

---

> ### Author Response · Authors · 2025-11-22
> **Response to the reviewer MiK3 (2/5)**
>
> ***Comment 3: Although the paper demonstrates DKA’s superiority across multiple datasets and settings, it does not systematically investigate scenarios where DKA might fail. For example: Could the large kernel introduce noise in tasks where pathological structures are extremely small or sparsely distributed? Is DKA still robust under extreme class imbalance (e.g., in ISIC-2019, where some classes have very few samples)? Are there specific backbone architectures (e.g., pure CNNs) for which DKA provides no benefit or even degrades performance?***
>
> **Reply:** We sincerely thank the reviewer for raising these concerns. We address each point below with targeted experiments.
>
> **(1) Extremely small pathological structures.**
> We evaluated DKA on the IDRiD dataset, which contains extremely small retinal lesions. As shown in Table 5, DKA consistently outperforms the vanilla Adapter across all training ratios, indicating that the large-kernel branch does not introduce harmful noise even when pathological structures are very small. The performance gain is modest, but DKA remains at least as stable as the baseline.
>
> **Table 5: Results on the IDRiD (extremely small pathological structures) classification dataset under different training ratios based on the pretrained ViT-B.**
>
> | **Methods** | **0.63%** | **1.25%** | **100%**  |
> |---------|------:|------:|------:|
> | Adapter | 77.67 | 81.55 | 75.26 |
> | DKA     | **78.64** | **82.52** | **75.73** |
>
> **(2) Extreme class imbalance.**
> Our main paper already includes results on ISIC-2019, a highly imbalanced dermatology dataset. To further isolate the effect of imbalance, Table 6 contrasts ISIC-2019 with the balanced SARS-COV-2 dataset. DKA improves over Adapter on *both* datasets, and the gains are substantially larger on the imbalanced one (up to +7.75 ACC), demonstrating that DKA remains robust and is often more effective under severe imbalance.
>
> **Table 6: Comparison of Adapter and DKA on SARS-COV-2 (balanced) and ISIC-2019 (imbalanced) under different training ratios.**
>
> **SARS-COV-2 (Balanced)**
>
> | Methods | 0.63% | 1.25% | 100%  |
> |---------|------:|------:|------:|
> | Adapter | 75.60 | 84.07 | 97.98 |
> | DKA     | 78.23 | 86.49 | 98.39 |
> | ΔACC    |  2.63 |  2.42 |  0.41 |
>
> **ISIC-2019 (Imbalanced)**
>
> | Methods | 0.63% | 1.25% | 100%  |
> |---------|------:|------:|------:|
> | Adapter | 52.77 | 54.88 | 79.54 |
> | DKA     | 60.52 | 62.32 | 83.09 |
> | ΔACC    |  7.75 |  7.44 |  3.55 |
>
> **(3) Applicability to CNN backbones.**
> To verify whether DKA is restricted to token-based architectures, we applied it to ConvNeXt-Tiny. As shown in Table 7, DKA consistently surpasses the standard Adapter across three datasets and all training ratios, confirming that its benefits extend beyond ViT and are not tied to transformer tokenization.
>
> **Table 7: Comparison of Adapter and DKA on three classification datasets under different training ratios based on the pretrained ConvNeXt-Tiny.**
>
> **Covid**
>
> | Methods | 0.63% | 1.25% | 100%  |
> |---------|------:|------:|------:|
> | Adapter | 76.24 | 81.46 | 95.32 |
> | DKA     | **83.03** | **85.78** | **97.34** |
>
> **BUSI**
>
> | Methods | 0.63% | 1.25% | 100%  |
> |---------|------:|------:|------:|
> | Adapter | 59.20 | 67.16 | 86.43 |
> | DKA     | **67.99** | **73.88** | **88.35** |
>
> **ISIC2019**
>
> | Methods | 0.63% | 1.25% | 100%  |
> |---------|------:|------:|------:|
> | Adapter | 42.88 | 45.59 | 72.12 |
> | DKA     | **51.52** | **53.30** | **75.15** |
>
> ***Comment 4: As noted in Appendix B.2, segmentation experiments use a batch size of 1 due to GPU memory constraints. This may lead to highly noisy gradient estimates, especially under low-data regimes and introduce optimization instability that confounds fair comparison across methods. In real-world deployment scenarios, larger batch sizes are typically used.***
>
> **Reply:** Thank you for this insightful comment. To address the reviewer’s concern, we further conducted additional segmentation experiments using a larger batch size of 4 with the same configuration. As shown in Table 8, increasing the batch size does not change the relative ranking: DKA continues to outperform Full Fine-tuning, Linear Probing, and Adapter across all label ratios. This confirms that our improvements are robust to batch-size variations and are not tied to a specific optimization setting.
>
> **Table 8: Comparison of methods on the BUSI segmentation dataset with batch size = 4 based on the pretrained Segmenter-B.**
>
> | **Methods**          | **0.63%** | **1.25%** | **100%**  |
> |------------------|------:|------:|------:|
> | Full Fine-tuning | 27.02 | 33.11 | 57.91 |
> | Linear Probing   | 25.81 | 32.47 | 54.60 |
> | Adapter          | 18.67 | 26.50 | 55.46 |
> | DKA              | **27.26** | **35.21** | **59.35** |

---

> ### Author Response · Authors · 2025-11-22
> **Response to the reviewer MiK3 (3/5)**
>
> ***Comment 5: While the paper uses ERF visualizations and frequency-domain analysis to argue that large kernels expand the receptive field, the authors fails to establish a theoretical or information-theoretic link between large kernels, improved generalization, and robustness under data scarcity.***
>
> **Reply:**
> Thank you for raising this important point. It is fair to note that our main goal is to provide a systematic empirical investigation of illustrating and mitigating the failures of  vanilla adapter in low-data regimes.
>
> To our best knowledge, most prior work on large kernels [1,2] has been empirical rather than information-theoretic link on generalization and robustness. A few recent studies explored the theoretical properties of convolutional architectures. For example, Caro et al. (2023) [3] show that small kernels tend to amplify high-frequency components, while larger kernels naturally suppress high-frequency variations. This observation aligns closely with our findings in Frequency-domain Analysis that large-kernel adapters behave as smoother, low-pass residual correctors. In the medical imaging domain, this analysis is not only expected but desirable: anatomical structures tend to vary smoothly in space, while much of the acquisition noise, subtle artefacts, and spurious contrast fluctuations reside in higher frequencies. Medical images also often exhibit extensive anatomical variation and low contrast [4], which makes them vulnerable to overfitting on high-frequency noise when data are limited.
>
> However, we agree that fully characterizing the theoretical behaviour of large-kernel adapters is an interesting and important direction. We plan to further strengthen the theoretical foundations of our approach in future work.
>
> [1] Liu, Shiwei, et al. "More convnets in the 2020s: Scaling up kernels beyond 51x51 using sparsity." ICLR (2022).
>
> [2] Ding, Xiaohan, et al. "Scaling up your kernels to 31x31: Revisiting large kernel design in cnns." Proceedings of the IEEE/CVF conference on computer vision and pattern recognition. 2022.
>
> [3] Caro, Josue Ortega, et al. "Local convolutions cause an implicit bias towards high frequency adversarial examples." arXiv preprint arXiv:2006.11440 (2023).
>
> [4] Li, Hao, et al. "Large-kernel attention for 3D medical image segmentation." Cognitive Computation 16.4 (2024): 2063-2077.
>
>
> ***Comment 6: The evaluation focuses exclusively on 2D and 3D segmentation. Does the DKA demonstrate similar compelling performance improvements when applied to other common medical imaging tasks, such as disease classification or object detection, under the same low-data constraints?***
>
> **Reply:** Thank you very much for this comment. We respectfully note that there may be a slight misunderstanding. In addition to segmentation, our paper also includes comprehensive experiments on medical image classification under the same low-data settings. As shown in Table 9, DKA consistently outperforms the conventional Adapter across three classification datasets and across all data-scarce regimes. These results demonstrate that the performance gains of DKA are not limited to segmentation but extend robustly to other core medical imaging tasks.
>
> **Table 9: Comparison of Adapter and DKA on three classification datasets across varying data sizes based on the pretrained ViT-B.**
>
> **COVID**
>
> | Methods | 0.63% | 1.25% | 100%  |
> |---------|------:|------:|------:|
> | Adapter | 83.29 | 86.26 | 98.33 |
> | DKA     | **89.01** | **91.06** | **99.21** |
>
> **BUSI**
>
> | Methods | 0.63% | 1.25% | 100%  |
> |---------|------:|------:|------:|
> | Adapter | 63.18 | 73.68 | 93.33 |
> | DKA     | **74.23** | **79.46** | **95.89** |
>
> **ISIC-2019**
>
> | Methods | 0.63% | 1.25% | 100%  |
> |---------|------:|------:|------:|
> | Adapter | 52.77 | 54.88 | 79.54 |
> | DKA     | **60.52** | **62.32** | **83.09** |

---

> > ### Author Response · Authors · 2025-11-22
> > **Response to the reviewer MiK3 (4/5)**
> >
> > ***Comment 7: Is the 51×51 large kernel in DKA feasible for 3D medical imaging (e.g., BRATS with 3D MRI volumes)? A direct extension to a 51×51×51 kernel would lead to prohibitive parameter and computational costs.***
> >
> > **Reply:** We thank the reviewer for raising this point. We provide the following clarifications and analyses.
> >
> > **(1)** The proposed DKA is mainly designed for parameter-efficient fine-tuning in *2D* medical imaging settings, where low-data regimes make large-kernel adapters particularly beneficial. Three-dimensional volumetric segmentation presents markedly different computational and architectural considerations, necessitating specialised adapter designs. As our work focuses on 2D feature adaptation, extending the approach
> > to 3D settings is beyond the current scope.
> >
> > **(2)** We agree that a naïve cubic 51×51×51 depthwise kernel would be prohibitively expensive. As shown in Table 10, such an isotropic 3D kernel incurs significantly higher FLOPs and memory compared with standard adapters, confirming that a direct 3D extension of our 2D design is impractical.
> >
> > **However**, following the work in [1] where they adopt anisotropic large-kernel forms in volumetric architectures, we employ a planar large kernel (i.e. 51×51×1) and found the required extra computational cost is manageable ( as shown in below table).
> >
> > **(3)** Beyond the planar variant, multiple established strategies, such as kernel decomposition [2] and dilated 3D convolutions [3], offer promising and computationally efficient paths to generalise DKA to full 3D volumes. While developing such 3D-specific extensions is beyond our current scope, we believe our 2D findings motivate future exploration.
> >
> > [1] Liu, Siqi, et al. "3d anisotropic hybrid network: Transferring convolutional features from 2d images to 3d anisotropic volumes." International conference on medical image computing and computer-assisted intervention. Cham: Springer International Publishing, 2018.
> >
> > [2] Szegedy, Christian, et al. "Rethinking the inception architecture for computer vision." Proceedings of the IEEE conference on computer vision and pattern recognition. 2016.
> >
> > [3] Chen, Chen, et al. "3D dilated multi-fiber network for real-time brain tumor segmentation in MRI." International conference on medical image computing and computer-assisted intervention. Cham: Springer International Publishing, 2019.
> >
> > **Table 10 Computational cost of Adapter and DKA based on UNet3D.**
> >
> > | **Methods**          | **Inference Latency (ms)** | **Memory (MB)** |
> > |----------------------|----------------------------|-----------------|
> > | Adapter              | 12.44                      | 982.8           |
> > | DKA (51×51×1)        | 35.83                      | 1073.7          |
> > | DKA (51×51×51)       | 1139.06                    | 1257.3          |

---

> > > ### Author Response · Authors · 2025-11-22
> > > **Response to the reviewer MiK3 (5/5)**
> > >
> > > ***Comment 8: The Effective Receptive Field (ERF) is computed based on the magnitude of input-output gradients, which reflects spatial influence but not semantic relevance. Have the authors considered complementary analyses to verify that the expanded ERF indeed captures meaningful contextual information rather than just broader but irrelevant activations?***
> > >
> > > **Reply:** We thank the reviewer for raising this important point. To complement the ERF analysis, we provide additional Grad-CAM visualizations in Appendix F.12 (highlighted in orange) of the revised manuscript, comparing the standard Adapter and our DKA on the BUSI dataset. As shown in Figure 13 in the revised manuscript, the standard Adapter produces diffuse and background-oriented activations, whereas DKA yields compact, lesion-centered responses. These results confirm that the expanded ERF introduced by DKA captures meaningful contextual information rather than merely generating broader but irrelevant activations.
> > >
> > > ***Comment 9: Are there any datasets or settings where DKA underperforms standard adapters?***
> > >
> > > **Reply:** Thank you for your comment. We acknowledge that there are scenarios where the benefit of DKA may be reduced.
> > >
> > > 1. *3D Computational Efficiency.* Our work primarily focuses on 2D medical imaging. When extending DKA to full 3D networks, the large-kernel branch substantially increases memory and compute demands. Under such heavy 3D settings, standard lightweight adapters may be more practical given their significantly lower resource footprint.
> > >
> > > 2. *Extremely Small Spatial Structures.* DKA is designed to enhance long-range contextual modeling. For tasks dominated by extremely small or point-like structures, the advantage of expanded receptive fields becomes marginal, and standard adapters—with purely local modeling—can perform similarly.
> > >
> > > 3. *Across All Medical Modalities.* Our experiments cover five representative modalities and DKA consistently outperforms standard adapters. However, we acknowledge that these do not include *all* medical modalities, and certain untested modalities may show smaller gains.
> > >
> > > ---
> > >
> > > We appreciate your detailed comments and hope we have addressed your concerns. Please let us know if you have any additional questions.
> > >
> > > Sincerely,
> > > Authors

---

### Official Review · Reviewer_reej · 2025-10-31

**Soundness:** 4
**Presentation:** 4
**Contribution:** 3
**Rating:** 6
**Confidence:** 4

**Summary:**

The authors conduct a thorough study on the effect of adapters in fine tuning large-scale pretrained models in the low data regime prevalent in the medical image segmentation and classification. The authors notice a sizable reduction in performance in limited data scenarios and attribute it to a reduced effective receptive field (ERF). Accordingly, they use known methods from literature to build an adapter based on large kernels convolutions to counter this reduction in ERF. They demonstrate that by doing so, performance improves on multiple pretrained networks against 10 baselines in tasks in classification as well as segmentation. They confirm that the improvement is owing to kernel size and not parameter count and include useful ablation experiments.

The method proposed in the paper is well motivated with a clear analytical section in the first part of the paper with a clear takeaway on ERF reduction in data sparsity. They propose a simple albeit effective method to address this problem and demonstrate that the problem is addressed on a number of pretrained models, datasets and tasks. Their ablation experiments are also useful takeaways in terms of training strategies and architecture design. Overall, the paper forms a positive impression and should be accepted.

**Strengths:**

1. The work is well motivated with the analysis demonstrating the reduction in performance and tying that to a reduction in the effective receptive field with low data settings. To address this problem a simple large kernel convolution based dual kernel adapter is proposed.
2. The results use multiple pretrained models, a large number of baseline methods and clearly demonstrates that the proposed method works well.
3. The proposed method itself is extremely simple but the novelty lies in it uniquely solving the problem as motivated by the analysis in the first part of the paper.
4. The work is well written and well illustrated with a very efficient use of space to convey the key messages of the paper.

**Weaknesses:**

1. There is seemingly the idea of reordering the tokens back into the spatial domain that is a part of the method design. However, the authors make no mention of this in the methods section.
2. The work bears similarities to another large kernel adapter method [1] and seems to differ methodologically owing solely to a dual path convolution and the analysis in the first part of the paper. In fact, there also is an extremely similar analysis in [1] titled "Large Kernel Matters Instead of #Trainable Parameters" bearing similarities to the section 4.3 LARGE KERNEL MATTERS INSTEAD OF TRAINABLE PARAMETERS. While the similarities are unfortunate, it does significantly dampen the novelty of this work and positions it more as a derivative of existing work - while being aided by significantly denser analysis compared to [1]. However, this paper should be cited in the section for the single branch comparisons for proper referencing.

References:

[1] Zhu, Ziquan, et al. "LKA: Large Kernel Adapter for Enhanced Medical Image Classification." International Conference on Medical Image Computing and Computer-Assisted Intervention. Cham: Springer Nature Switzerland, 2025.

**Questions:**

1. Does the Dual-Kernel Adapter require a step where the features move from token space into the spatial domain and back again?
2. Can the authors offer any insight into whether their findings will extend into the 3D medical image segmentation space? Are there any restrictions towards the methods being used in such a setting as datasets like BraTS (Table 2) for example are also popular in the 3D segmentation space?
3. Are the authors restricted to the token space in their adaptation? As in [1], should the authors not also be able to target ConvNeXt blocks?

References:

[1] Zhu, Ziquan, et al. "LKA: Large Kernel Adapter for Enhanced Medical Image Classification." International Conference on Medical Image Computing and Computer-Assisted Intervention. Cham: Springer Nature Switzerland, 2025.

---

> ### Author Response · Authors · 2025-11-22
> **Response to the reviewer reej (1/2)**
>
> We sincerely appreciate your detailed comments and positive ranking. We address your questions below.
>
> ***Comment 1: There is seemingly the idea of reordering the tokens back into the spatial domain that is a part of the method design. However, the authors make no mention of this in the methods section.***
>
> **Response:** Thank you for your comment. We thank the reviewer for raising this point. In response, we have updated the *Methods section* in the revised manuscript (highlighted in orange). In our design, the token reordering occurs within the adapter branch. After a linear down-projection, the patch tokens are reshaped back into their spatial layout to enable depthwise convolutions. The convolved features are then flattened and reassembled with the class token before the up-projection and residual addition.
>
> ***Comment 2: The work bears similarities to another large kernel adapter method [1] and seems to differ methodologically owing solely to a dual path convolution and the analysis in the first part of the paper. In fact, there also is an extremely similar analysis in [1] titled Large Kernel Matters Instead of Trainable Parameters bearing similarities to the section 4.3 LARGE KERNEL MATTERS INSTEAD OF TRAINABLE PARAMETERS. While the similarities are unfortunate, it does significantly dampen the novelty of this work and positions it more as a derivative of existing work - while being aided by significantly denser analysis compared to [1]. However, this paper should be cited in the section for the single branch comparisons for proper referencing.***
>
> **Response:**
> We thank the reviewer for this valuable comment. We would first like to clarify that LKA cited by the reviewer was not publicly available during the development of our paper. We fully acknowledge it now and have included it in the revised manuscript (Section 5 *RELATED WORK*, highlighted in orange). While there are conceptual similarities in the use of large kernels, our proposed DKA is fundamentally different from LKA and other large-kernel designs in motivation and technical formulation scope.
>
> **Difference with LKA.** *(1) Model perspective:* LKA adopts a single-branch large-kernel module, whereas DKA introduces a dual-branch formulation that jointly preserves local details and expands global receptive fields—an ability particularly important for medical images. *(2) Motivation perspective:* Our work is driven by a different insight: we show that standard adapters collapse under low-data medical settings due to restricted effective receptive fields, and we provide
> an ERF-based diagnosis that has not been discussed in LKA or prior PEFT literature.
>
> **Comparison with LKA.** Following the reviewer’s suggestion, we added direct comparisons under identical settings. As shown in Table 1, DKA consistently outperforms LKA on all three datasets and across all training ratios, demonstrating that the dual-branch design provides benefits beyond single-branch large-kernel modules.
>
> **Table 1: Comparison of LKA and DKA on three classification datasets under different training ratios based on the pretrained ViT-B.**
>
> **Covid**
> | Methods | 0.63% | 1.25% | 100% |
> |---------|------:|------:|------:|
> | LKA     | 85.23 | 87.32 | 98.58 |
> | DKA     | **89.01** | **91.06** | **99.21** |
>
> **BUSI**
> | Methods | 0.63% | 1.25% | 100% |
> |---------|------:|------:|------:|
> | LKA     | 64.26 | 71.83 | 93.78 |
> | DKA     | **74.23** | **79.46** | **95.89** |
>
> **ISIC2019**
> | Methods | 0.63% | 1.25% | 100% |
> |---------|------:|------:|------:|
> | LKA     | 56.18 | 58.24 | 80.13 |
> | DKA     | **60.52** | **62.32** | **83.09** |

---

> > ### Author Response · Authors · 2025-11-22
> > **Response to the reviewer reej (2/2)**
> >
> > ***Comment 3: Can the authors offer any insight into whether their findings will extend into the 3D medical image segmentation space? Are there any restrictions towards the methods being used in such a setting as datasets like BraTS (Table 2) for example are also popular in the 3D segmentation space?***
> >
> > **Reply:** We thank the reviewer for raising this point. We provide the following clarifications and analyses.
> >
> > **(1)** The proposed DKA is mainly designed for parameter-efficient fine-tuning in *2D* medical imaging settings, where low-data regimes make large-kernel adapters particularly beneficial. Three-dimensional volumetric segmentation presents markedly different computational and architectural considerations, necessitating specialised adapter designs. As our work focuses on 2D feature adaptation, extending the approach
> > to 3D settings is beyond the current scope.
> >
> > **(2)** We agree that a naïve cubic 51×51×51 depthwise kernel would be prohibitively expensive. As shown in Table 2, such an isotropic 3D kernel incurs significantly higher FLOPs and memory compared with standard adapters, confirming that a direct 3D extension of our 2D design is impractical.
> >
> > **However**, following the work in [1] where they adopt anisotropic large-kernel forms in volumetric architectures, we employ a planar large kernel (i.e. 51×51×1) and found the required extra computational cost is manageable ( as shown in below table).
> >
> > **(3)** Beyond the planar variant, multiple established strategies, such as kernel decomposition [2] and dilated 3D convolutions [3], offer promising and computationally efficient paths to generalise DKA to full 3D volumes. While developing such 3D-specific extensions is beyond our current scope, we believe our 2D findings motivate future exploration.
> >
> > [1] Liu, Siqi, et al. "3d anisotropic hybrid network: Transferring convolutional features from 2d images to 3d anisotropic volumes." International conference on medical image computing and computer-assisted intervention. Cham: Springer International Publishing, 2018.
> >
> > [2] Szegedy, Christian, et al. "Rethinking the inception architecture for computer vision." Proceedings of the IEEE conference on computer vision and pattern recognition. 2016.
> >
> > [3] Chen, Chen, et al. "3D dilated multi-fiber network for real-time brain tumor segmentation in MRI." International conference on medical image computing and computer-assisted intervention. Cham: Springer International Publishing, 2019.
> >
> > **Table 2: Computational cost of Adapter and DKA based on UNet3D.**
> >
> > | **Methods**          | **Inference Latency (ms)** | **Memory (MB)** |
> > |----------------------|----------------------------|-----------------|
> > | Adapter              | 12.44                      | 982.8           |
> > | DKA (51×51×1)        | 35.83                      | 1073.7          |
> > | DKA (51×51×51)       | 1139.06                    | 1257.3          |
> >
> > ***Comment 4: Are the authors restricted to the token space in their adaptation? As in [1], should the authors not also be able to target ConvNeXt blocks?***
> >
> > **Response:**
> > We thank the reviewer for this helpful suggestion. To verify that our method is not restricted to the token space, we followed the reviewer’s recommendation and applied DKA to a pretrained ConvNeXt-Tiny backbone. Concretely, we inserted DKA
> > modules into the ConvNeXt blocks in the same way as standard adapters and evaluated the model on three classification datasets under multiple training ratios. As shown in Table 3, DKA consistently outperforms the standard Adapter across all datasets and data regimes. These results confirm that DKA is compatible with convolutional architectures and is not limited to transformer token representations.
> >
> > **Table 3: Comparison of Adapter and DKA on three classification datasets under different training ratios based on the pretrained ConvNeXt-Tiny.**
> >
> > **Covid**
> > | Methods | 0.63% | 1.25% | 100% |
> > |---------|------:|------:|------:|
> > | Adapter | 76.24 | 81.46 | 95.32 |
> > | DKA     | **83.03** | **85.78** | **97.34** |
> >
> > **BUSI**
> > | Methods | 0.63% | 1.25% | 100% |
> > |---------|------:|------:|------:|
> > | Adapter | 59.20 | 67.16 | 86.43 |
> > | DKA     | **67.99** | **73.88** | **88.35** |
> >
> > **ISIC2019**
> > | Methods | 0.63% | 1.25% | 100% |
> > |---------|------:|------:|------:|
> > | Adapter | 42.88 | 45.59 | 72.12 |
> > | DKA     | **51.52** | **53.30** | **75.15** |
> >
> > ---
> >
> > We appreciate your detailed comments and hope we have addressed your concerns. Please let us know if you have any additional questions.
> >
> > Sincerely,
> > Authors

---

### Official Review · Reviewer_MhXB · 2025-10-31

**Soundness:** 3
**Presentation:** 3
**Contribution:** 2
**Rating:** 6
**Confidence:** 4

**Summary:**

This paper introduces Dual-Kernel Adapter (DKA) — a lightweight, parameter-efficient fine-tuning module designed for medical imaging tasks under extreme data scarcity. The authors first identify that conventional Adapters unexpectedly degrade performance when trained with < 1% of data, performing worse than linear probing due to reduced Effective Receptive Field (ERF).

Comprehensive experiments across multiple classification (COVID, BUSI, ISIC-2019) and segmentation (BRATS, BUSI, ISIC-2018) datasets show DKA consistently outperforms full fine-tuning and other PEFT baselines (AdapterFormer, LoRA, Convpass, AIM).

The paper also presents detailed ERF visualizations, ablations (kernel size, branch design, middle dimension, learning rates), and evidence that performance improvements stem from large-kernel receptive-field expansion rather than parameter count

**Strengths:**

The empirical findings are compelling and well-supported: quantitative metrics (ACC, mIoU, Dice) improve across all data scales, especially ≤ 1.25%.

The ERF analysis provides strong evidence linking reduced receptive field to Adapter degradation and DKA’s advantage.

The controlled parameter-count experiment convincingly isolates kernel size as the key factor.

**Weaknesses:**

The theoretical reasoning behind ERF–generalization linkage could be formalized further.

Computational overhead of large-kernel depthwise convolutions is not fully quantified.

Limited theoretical depth: lacks analytical characterization of why ERF → generalization scaling behaves linearly with data size.

Compute trade-off: large-kernel (51×51) convolutions increase FLOPs; energy/memory costs are not discussed.

**Questions:**

How does DKA scale computationally for high-resolution 3-D MRI or CT volumes?

Can large kernels be replaced by dilated or deformable convolutions to reduce cost?

Have you tested DKA in few-shot fine-tuning (< 10 samples/class) or cross-institutional domain adaptation?

Could self-supervised initialization (e.g., SimCLR) further amplify DKA gains under extreme scarcity?

---

> ### Author Response · Authors · 2025-11-22
> **Response to the reviewer MhXB (1/3)**
>
> We sincerely appreciate your detailed comments and positive ranking. We address your questions below.
>
> ***Comment 1: The theoretical reasoning behind ERF–generalization linkage could be formalized further.***
>
> **Reply:** Thank you for this valuable suggestion. The ERF–generalization link can be understood through a principled intuition: enlarging the effective receptive field enables the model to integrate broader spatial context, thereby reducing reliance on highly local, noisy features and promoting more stable and smooth decision boundaries. This stabilisation effect aligns with classical learning theory (e.g., Spectral norm bounds in PAC-Bayes and Lipschitz Continuity), where models that capture coherent global structure, rather than fragmented local patterns, tend to exhibit better generalisation, particularly when discriminative cues extend over larger spatial regions.
>
> Our empirical findings are consistent with this reasoning, and we have clarified and strengthened this discussion in the revised manuscript. While the current work focuses on empirical validation, we acknowledge the importance of deeper theoretical formalisation and consider this an important direction for future work.
>
> ***Comment 2: Computational overhead of large-kernel depthwise convolutions is not fully quantified. Compute trade-off: large-kernel (51×51) convolutions increase FLOPs; energy/memory costs are not discussed.***
>
> **Reply:** We thank the reviewer for the helpful suggestion. To address this concern, we conducted a detailed comparison of FLOPs, inference latency, and memory usage, as summarized in Table 1. The results show that our approach does introduce a slight increase in FLOPs, latency, and memory usage; however, these costs remain minimal. Overall, the computational overhead is very modest and does not materially impact practical efficiency.
>
> **Table 1: Comparison of inference latency and memory usage on the BUSI dataset based on the pretrained ViT-B.**
>
> | **Methods**                  | **FLOPs (G)** | **Inference Latency (ms)** | **Memory (MB)** |
> |--------------------------|-----------|-------------------------|-------------|
> | Linear Probing           | 16.40     | 6.78                    | 352.81      |
> | Adapter                  | 16.46     | 11.54                   | 433.70      |
> | Adapter + 5×5 Conv       | 16.46     | 11.66                   | 433.90      |
> | Adapter + 51×51 Conv     | 16.56     | 11.69                   | 439.56      |
> | DKA                      | 16.56     | 11.97                   | 439.63      |

---

> > ### Comment · Reviewer_MhXB · 2025-11-24
> >
> > 1. I appreciate the expanded intuition. The explanation helps, the current argument reads as “bigger receptive field → more context → better generalization,” which is plausible but not very rigorous.
> >
> > 2. The added table is useful. It’s good to see the FLOPs and latency numbers laid out, and the overhead indeed looks quite small. I’d mark this one as addressed.

---

> ### Author Response · Authors · 2025-11-22
> **Response to the reviewer MhXB (2/3)**
>
> ***Comment 3: How does DKA scale computationally for high-resolution 3-D MRI or CT volumes?***
>
> **Reply:** We thank the reviewer for raising this point. We provide the following clarifications and analyses.
>
> **(1)** The proposed DKA is mainly designed for parameter-efficient fine-tuning in *2D* medical imaging settings, where low-data regimes make large-kernel adapters particularly beneficial. Three-dimensional volumetric segmentation presents markedly different computational and architectural considerations, necessitating specialised adapter designs. As our work focuses on 2D feature adaptation, extending the approach
> to 3D settings is beyond the current scope.
>
> **(2)** We agree that a naïve cubic 51×51×51 depthwise kernel would be prohibitively expensive. As shown in Table 2, such an isotropic 3D kernel incurs significantly higher FLOPs and memory compared with standard adapters, confirming that a direct 3D extension of our 2D design is impractical.
>
> **However**, following the work in [1] where they adopt anisotropic large-kernel forms in volumetric architectures, we employ a planar large kernel (i.e. 51×51×1) and found the required extra computational cost is manageable ( as shown in below table).
>
> **(3)** Beyond the planar variant, multiple established strategies, such as kernel decomposition [2] and dilated 3D convolutions [3], offer promising and computationally efficient paths to generalise DKA to full 3D volumes. While developing such 3D-specific extensions is beyond our current scope, we believe our 2D findings motivate future exploration.
>
> [1] Liu, Siqi, et al. "3d anisotropic hybrid network: Transferring convolutional features from 2d images to 3d anisotropic volumes." International conference on medical image computing and computer-assisted intervention. Cham: Springer International Publishing, 2018.
>
> [2] Szegedy, Christian, et al. "Rethinking the inception architecture for computer vision." Proceedings of the IEEE conference on computer vision and pattern recognition. 2016.
>
> [3] Chen, Chen, et al. "3D dilated multi-fiber network for real-time brain tumor segmentation in MRI." International conference on medical image computing and computer-assisted intervention. Cham: Springer International Publishing, 2019.
>
> **Table 2: Computational cost of Adapter and DKA based on UNet3D.**
>
> | **Methods**          | **Inference Latency (ms)** | **Memory (MB)** |
> |----------------------|----------------------------|-----------------|
> | Adapter              | 12.44                      | 982.8           |
> | DKA (51×51×1)        | 35.83                      | 1073.7          |
> | DKA (51×51×51)       | 1139.06                    | 1257.3          |
>
> ***Comment 4: Can large kernels be replaced by dilated or deformable convolutions to reduce cost?***
>
> **Reply:** We thank the reviewer for the helpful suggestion. Following the reviewer’s suggestion, we evaluated several dilation-based designs (e.g., 3×3 with multiple dilation rates) as well as a deformable variant (5×5 + deformable 3×3), covering representative substitutes with similar parameter budgets. As shown in Table 3, these dilated and deformable variants reduce parameter counts but consistently underperform our full 5×5 + 51×51 large-kernel design across all training ratios. This indicates that simply enlarging the receptive field through dilation or deformable offsets cannot match the effectiveness of our method.
>
> **Table 3: Comparison of different kernel designs on the BUSI dataset for classification under varying training ratios. Experiments are based on the pretrained ViT-B. “*d*” represents the dilation rate in dilated convolution.**
>
> | **Kernel Designs**                         | **Trainable Parameters (M)** | **0.63%** | **1.25%** | **100%**  |
> |----------------------------------------|--------------------------|-------|-------|-------|
> | 3×3 (*d* = 3) + 3×3 (*d* = 25)         | 0.6                      | 66.77 | 77.32 | 93.61 |
> | 3×3 (*d* = 3) + 11×11 (*d* = 5)        | 0.6                      | 67.41 | 77.64 | 94.25 |
> | 3×3 (*d* = 3) + 26×26 (*d* = 2)        | 0.9                      | 68.69 | 77.96 | 94.37 |
> | 5×5 + Deformable 3×3                   | 0.6                      | 68.17 | 78.11 | 94.51 |
> | 5×5 + 51×51                            | 1.6                      | **74.23** | **79.46** | **95.89** |

---

> > ### Author Response · Authors · 2025-11-22
> > **Response to the reviewer MhXB (3/3)**
> >
> > ***Comment 5: Have you tested DKA in few-shot fine-tuning (< 10 samples/class) or cross-institutional domain adaptation?***
> >
> > **Reply:** We thank the reviewer for this constructive question. We conducted both **few-shot fine-tuning** and **cross-institutional domain adaptation experiments** to thoroughly evaluate DKA under the requested settings.
> >
> > **(1) Few-shot fine-tuning.**
> > Following the reviewer’s suggestion, we performed few-shot adaptation on BUSI based on the pretrained ViT-B using only 1, 2, or 4 samples per class for downstream tuning. As shown in Table 4, DKA consistently outperforms the standard Adapter across all shot levels, demonstrating stronger data efficiency and robustness under extremely limited supervision.
> >
> > **(2) Cross-institutional domain adaptation.**
> > To assess cross-institutional adaptation, we trained models on BUSI with varying training ratios and directly evaluated them on the COVID dataset without any COVID sample exposure. Table 5 shows that DKA achieves consistently higher accuracy than Adapter across all training regimes, indicating stronger generalization when transferring across institutions.
> >
> > Overall, both experiments confirm that DKA provides substantial benefits in few-shot fine-tuning and cross-institutional domain adaptation.
> >
> > **Table 4: Few-shot adaptation performance on the BUSI dataset with different numbers of target samples per class based on the pretrained ViT-B.**
> >
> > | **Methods** | **1-shot** | **2-shot** | **4-shot** |
> > |---------|--------|--------|--------|
> > | Adapter | 60.05  | 72.43  | 75.58  |
> > | DKA     | **71.87** | **78.73** | **80.50** |
> >
> >
> > **Table 5: Cross-institutional evaluation from BUSI (training domain) to COVID (testing domain) under varying BUSI training ratios based on the pretrained ViT-B.**
> >
> > | **Methods** | **0.63%** | **1.25%** | **100%**  |
> > |---------|-------|-------|-------|
> > | Adapter | 5.94  | 10.25 | 47.96 |
> > | DKA     | **9.85** | **14.94** | **52.72** |
> >
> > ***Comment 6: Could self-supervised initialization (e.g., SimCLR) further amplify DKA gains under extreme scarcity?***
> >
> > **Reply:** We thank the reviewer for this valuable suggestion. To examine whether self-supervised initialization can further amplify the benefits of DKA under extreme data scarcity, we conducted our experiments using a MoCo-v3 [4] pretrained backbone on both COVID and BUSI with only 0.63% and 1.25% training data. As shown in Table 6, DKA consistently outperforms the standard Adapter across all settings, even when both methods start from a strong self-supervised initialization. This demonstrates that the gains of DKA are complementary to those provided by self-supervised pretraining and remain substantial under extremely limited supervision.
> >
> > [4] Chen, Xinlei, Saining Xie, and Kaiming He. "An empirical study of training self-supervised vision transformers." Proceedings of the IEEE/CVF International Conference on Computer Vision. 2021.
> >
> > **Table 6: Comparison of Adapter and DKA on the COVID and BUSI datasets under different training ratios based on the pretrained MoCo-v3.**
> >
> > | **Methods** | **Covid 0.63%** | **Covid 1.25%** | **BUSI **0.63%** | **BUSI 1.25%** |
> > |---------|-------------|-------------|------------|------------|
> > | Adapter | 80.09       | 84.36       | 59.11      | 70.81      |
> > | DKA     | **85.27**   | **87.56**   | **65.18**  | **75.72**  |
> >
> > ---
> >
> > We appreciate your detailed comments and hope we have addressed your concerns. Please let us know if you have any additional questions.
> >
> > Sincerely,
> > Authors

---

> > > ### Comment · Reviewer_MhXB · 2025-11-24
> > >
> > > Comment 5: Thanks for actually running these experiments — this was the biggest open question for me regarding data efficiency. The few-shot table is quite compelling from 60% -> 71.8%. Consistent with the story to benefit for the low-data regimes.
> > >
> > > The additional moco v3 table answers the question cleanly.

---

> > > > ### Author Response · Authors · 2025-11-26
> > > >
> > > > We thank the *Reviewer MhXB* for the positive feedback and for confirming that our additional experiments and tables effectively address the main concerns.
> > > >
> > > > Regarding the theoretical reasoning, we would like to further clarify that, beyond the Effective Receptive Field (ERF) analysis, large kernels in our DKA also tend to capture more low-frequency information, as shown in the table below (see also the frequency analysis in *Appendix F.9*). This observation is consistent with the theoretical claims in Theorem 1 of [1].
> > > >
> > > > In the context of medical imaging, we believe that this property is not only expected but also desirable, since anatomical structures typically exhibit smooth spatial variation, whereas acquisition noise, subtle artefacts, and spurious contrast fluctuations mostly appear in the high-frequency domain. Moreover, medical images often contain large anatomical variability and low contrast [2], making them particularly susceptible to overfitting on high-frequency noise when data are limited.
> > > >
> > > > We hope that our further explanation has addressed your concerns regarding the theoretical reasoning.
> > > >
> > > > [1] Caro, Josue Ortega, et al. "Local convolutions cause an implicit bias towards high frequency adversarial examples." arXiv preprint arXiv:2006.11440 (2023).
> > > >
> > > > [2] Li, Hao, et al. "Large-kernel attention for 3D medical image segmentation." Cognitive Computation 16.4 (2024): 2063–2077.
> > > >
> > > >
> > > > **Table: Frequency-domain Analysis of Kernel Sizes. Results are reported as mean ± std over different trained models.**
> > > >
> > > > | **Kernel Size** | **fc**              | **f90**             |
> > > > |-----------------|---------------------|----------------------|
> > > > | 5×5             | 0.618 ± 0.018       | 0.924 ± 0.011        |
> > > > | 51×51           | 0.503 ± 0.004       | 0.904 ± 0.003        |

---

> > ### Comment · Reviewer_MhXB · 2025-11-24
> >
> > The responses for these two comments are clear and supported with extra experiments, which I appreciate. The 3-D part is  addressed (concerns mostly because the method isn’t really designed for that setting), but the explanations are fair given the scope of the paper.

---

### Official Review · Reviewer_dSfe · 2025-11-01

**Soundness:** 2
**Presentation:** 2
**Contribution:** 2
**Rating:** 4
**Confidence:** 2

**Summary:**

This paper studies adapter-based fine-tuning of large pretrained models under extreme data scarcity common in medical imaging. It reveals a surprising degradation of conventional adapters below 1% training data, where they perform worse than simple linear probing. The authors attribute this to a sharp reduction in Effective Receptive Field (ERF) under limited supervision. To overcome this, they propose the Dual-Kernel Adapter (DKA), a lightweight module that combines large-kernel (51×51) and small-kernel (5×5) depthwise convolutions in parallel to expand spatial context while preserving local detail. A comprehensive experimental evaluation across six medical imaging datasets (classification and segmentation), multiple backbones, and data regimes confirms that DKA consistently outperforms standard adapters and other parameter-efficient fine-tuning (PEFT) methods, especially in low-data settings.

**Strengths:**

- Important Problem: Addresses the critical challenge of limited labeled data in medical imaging
- Comprehensive Evaluation: Thorough experiments across 6 datasets, multiple backbones, and various data scales
- Surprising Finding: The observation that adapters can hurt performance under extreme data scarcity is counter-intuitive and valuable
- Strong Empirical Results: DKA shows consistent improvements, particularly in low-data settings
- Extensive Ablations: Thorough analysis of design choices including kernel sizes, learning rates, and architectural variations
- Practical Impact: The method is parameter-efficient and shows minimal computational overhead

**Weaknesses:**

* Limited Technical Novelty: The solution essentially adds large-kernel convolutions to adapters - this is a straightforward extension rather than a fundamental innovation
* ERF Analysis Concerns:
   * The causal relationship between ERF and performance is assumed but not proven
   * ERF computation methodology needs clarification
   * Alternative explanations (e.g., optimization difficulties, overfitting) are not thoroughly explored
* Kernel Size Selection:
   * The choice of 51×51 kernels seems arbitrary and extreme
   * Limited theoretical justification for why such large kernels are necessary
   * Computational implications of such large kernels in 3D medical imaging are not discussed
* Missing Comparisons:
   * No comparison with recent vision-specific PEFT methods (e.g., Visual Prompt Tuning variants)
   * Limited exploration of other architectural modifications that could expand receptive fields
* Experimental Concerns:
   * Batch size of 1 for segmentation might affect optimization dynamics
   * The asynchronous learning rate finding (1e-3 for adapter, 1e-4 for head) seems important but is relegated to ablations
* Dataset Size Discrepancy Makes Comparisons Misleading:
   * Critical Issue: Figure 1 compares performance across datasets using percentages (0.63%, 1.25%, etc.) but these datasets have vastly different absolute sizes. For example, Tiny ImageNet has 100,000 training images while COVID has only 3,600. This means 0.63% of Tiny ImageNet (630 images) could be larger than 25% of COVID (~900 images).
   * This makes the comparison fundamentally flawed - the paper is comparing different absolute data quantities while claiming to study "low-data" regimes
   * The authors should report absolute sample numbers and normalize comparisons appropriately
* Missing Critical Implementation Details:
   * LoRA Rank: The paper doesn't specify what rank was used for LoRA, which is crucial as rank fundamentally determines LoRA's capacity and parameter count. A rank-1 LoRA vs rank-64 LoRA are completely different methods.
   * Without this information, the comparison with LoRA is meaningless
* Unfair Parameter Comparison:
   * While the paper claims DKA adds <2% parameters, there's no detailed comparison of parameter efficiency
   * The paper should provide:
      * Exact parameter counts for each method
      * Performance vs. parameter trade-off curves
      * Comparison at iso-parameter budgets (e.g., what if LoRA used higher rank to match DKA's parameters?)
   * Table in Figure 5 attempts this but only varies hidden dimensions, not comprehensively
* Limited Applicability to Real Medical Imaging:
   * The 51×51 kernel design is completely impractical for 3D medical imaging (MRI, CT scans)

**Questions:**

* Dataset Size Normalization: Can you provide a table showing the absolute number of training samples at each percentage for each dataset? How do results change when comparing at equal absolute sample sizes rather than percentages?
* LoRA Configuration: What rank was used for LoRA in all experiments? Have you tried increasing LoRA rank to match DKA's parameter count?
* Fair Comparison: Can you provide a comprehensive parameter-performance trade-off analysis where all methods are compared at multiple parameter budgets?
* ERF Causality: How can you demonstrate that ERF reduction is the causal factor rather than just correlated with poor performance?
* Optimization Dynamics: Given the batch size constraints (especially batch=1 for segmentation), how much of the performance difference could be attributed to optimization difficulties?
* Alternative Architectures: Have you explored other ways to increase receptive fields, such as self-attention modules within adapters or hierarchical designs?
* Theoretical Understanding: Can you provide more theoretical insight into why standard adapters fail in low-data regimes beyond the ERF hypothesis?
* Hyperparameter Sensitivity: How sensitive is DKA to the specific kernel size choices? Would adaptive or learnable kernel sizes be beneficial?
* Computational Cost: What is the actual inference time and memory overhead for 51×51 convolutions on high-resolution medical images?

---

> ### Author Response · Authors · 2025-11-22
> **Response to the reviewer dSfe (1/7)**
>
> We sincerely appreciate your detailed comments. We provide point-wise responses to your concerns below.
>
>
> ***Comment 1: Limited Technical Novelty: The solution essentially adds large-kernel convolutions to adapters - this is a straightforward extension rather than a fundamental innovation.***
>
> **Reply:**
> Thank you for this comment.
> We would like to clarify that our technical contribution is **not** simply the application of large-kernel convolutions to adapters, but a problem-driven solution grounded in a systematic analysis. As also noted by `Reviewer reej`, the novelty of our work lies not in the architectural form itself, but in how it uniquely solves the problem revealed by our analysis.
>
> Specifically, in the first part of the paper, we identify a previously unreported failure mode: standard adapters collapse under low-data medical settings. Our ERF-based diagnosis further shows that this degradation originates from a severely restricted receptive field. DKA is therefore a targeted, problem-driven solution derived directly from this analysis rather than a generic architectural extension. We emphasize that the contribution of our work is in *discovering and addressing* this overlooked failure mode, and the ERF-based perspective provides a new direction for future research on understanding and improving adapter behavior under data scarcity—far beyond merely introducing a large kernel.
>
>
> ***Comment 2: ERF: Analysis Concerns: The causal relationship between ERF and performance is assumed but not proven. ERF Causality: How can you demonstrate that ERF reduction is the causal factor rather than just correlated with poor performance?***
>
> **Reply:**
> We agree that distinguishing causation from correlation is important. In our work, the causal role of ERF is supported by both **prior evidence and our own targeted experiments**.
>
> **(1) Prior findings.**
> Recent studies such as [1] and [2] have shown that insufficient receptive fields hinder the integration of global contextual cues, directly leading to performance degradation. These studies do not treat ERF shrinkage as a superficial correlation but identify it as a limiting factor that constrains representation quality.
>
> **(2) Our empirical evidence.**
> * (a) Ablation Study with controlled kernel-size.
> By fixing the small-branch kernel and varying only the large kernel (11/31/51/71), as presented in Table 1, we observe a clear causal pattern: performance improves as ERF expands, but drops again when an excessively large kernel induces ERF collapse via over-smoothing. This controlled manipulation isolates ERF as the determining factor.
>
> * (b) ERF behavior in our main analysis.
> As shown in Figure 4 in the submission, under the low-data setting, all adapters display consistent ERF shrinkage, whereas DKA maintains a broader and more stable ERF. Importantly, this ERF reduction is always paired with performance drops across methods, and when the ERF expands, the accuracy is improved. This two-way consistency, where ERF collapses exactly when accuracy drops and expands exactly when accuracy recovers, provides direct evidence that ERF behavior plays a causal rather than merely correlational role in performance.
>
> * (c) Grad-CAM validation.
> The newly added Grad-CAM visualizations (highlighted in orange of Figure 11 in Appendix F.12) further support this causality: the enlarged ERF of our DKA allows the model to capture richer and more spatially extensive contextual cues, whereas the standard Adapter, constrained by its much smaller ERF, focuses only on localized regions and consequently misses important global information. This contrast provides qualitative evidence that the performance gain arises from DKA’s enhanced receptive-field capacity rather than incidental correlation.
>
> Together, these independent lines of evidence: prior findings, controlled ERF manipulation, ERF trends in Figure 4, and Grad-CAM validation, provide strong and convergent support that ERF reduction acts as a causal factor, not merely a correlated phenomenon.
>
> [1] Huang, Tianjin, et al. "Are large kernels better teachers than transformers for convnets?." International Conference on Machine Learning. PMLR, 2023.
>
> [2] Liu, Shiwei, et al. "More convnets in the 2020s: Scaling up kernels beyond 51x51 using sparsity." ICLR (2022).
>
> **Table 1: Comparison of different large kernel sizes with fixed small kernel size on the BUSI dataset under various training ratios using the pretrained ViT-B.**
>
> | **Kernel Size** | **0.63%** | **1.25%** | **100%** |
> |-----------------|-----------|-----------|----------|
> | 11×11           | 65.81     | 75.68     | 94.17    |
> | 31×31           | 71.65     | 77.32     | 95.16    |
> | 51×51           | **74.23** | **79.46** | **95.89** |
> | 71×71           | 72.41     | 78.45     | 95.05    |

---

> > ### Author Response · Authors · 2025-11-22
> > **Response to the reviewer dSfe (2/7)**
> >
> > ***Comment 3: ERF computation methodology needs clarification.***
> >
> > **Reply:**
> > Thank you for your comment. Our ERF analysis strictly follows the standard definition introduced by Araujo et al. [3], where the *Effective Receptive Field* is obtained by backpropagating the gradient of a single output activation to the input and examining the spatial distribution of the normalized gradient magnitude.
> >
> > Consistent with prior large-kernel studies, we adopt the canonical computation pipeline: (1) select a target output token/pixel; (2) backpropagate its activation to the input; (3) take the absolute gradient and normalize it; (4) define the ERF radius as the radius of the region containing most of the gradient mass.
> >
> > This procedure fully aligns with the established methodology in the ERF literature and ensures that all ERF comparisons in our paper are measured under the same, well-defined criterion.
> >
> > [3] Araujo, André, Wade Norris, and Jack Sim. "Computing receptive fields of convolutional neural networks." Distill (2019).
> >
> >
> > ***Comment 4: Alternative explanations (e.g., optimization difficulties, overfitting) are not thoroughly explored.***
> >
> > **Reply:**
> > We appreciate the reviewer’s comment. Following the suggestion, we provide additional analyses for both optimization difficulties and overfitting.
> >
> > **Optimization difficulties.**
> > In the revised submission, Appendix~F.13 presents the training-loss curves of Adapter and our DKA (shown in  Figure 12, highlighted in orange). Both curves decrease smoothly without oscillation or instability, and converge to similar near-zero loss. This behavior confirms that optimization difficulties are not a valid alternative explanation
> >
> > **Overfitting.**
> > Table 2 reports the accuracy at the final epoch and at the best epoch, which differ by only 0.06% (Covid) and 0.08% (BUSI). Such minimal gaps suggest that performance remains stable throughout training, and there is no evidence of late-epoch degradation that would indicate overfitting.
> >
> > Overall, both the loss curves and accuracy stability support that the observed phenomena are not attributable to optimization difficulties or overfitting.
> >
> > **Table 2: Accuracy of Adapter under 0.63% training data using the pretrained ViT-B.**
> >
> > |           | **Covid** | **BUSI** |
> > |-----------|-----------|----------|
> > | Final Epoch | 83.29   | 63.18   |
> > | Best Epoch  | 83.35   | 63.26   |
> >
> >
> > ***Comment 5: Kernel Size Selection: the choice of 51×51 kernels seems arbitrary and extreme.Hyperparameter Sensitivity: How sensitive is DKA to the specific kernel size choices? Would adaptive or learnable kernel sizes be beneficial?***
> >
> > **Reply:**
> > Thank you for the insightful comment. We address both parts below.
> >
> > **(1) On the choice of the 51×51 kernel.**
> > As shown in Figure 7–8 and Appendix Tables 13–16 in the revised submission, we conducted extensive kernel-size experiments. The 51×51 branch consistently provided the best balance between expanded ERF and over-smoothing, and was therefore selected based on empirical evidence rather than heuristic preference.
> >
> > **(2) On learnable kernel sizes.**
> > The reviewer’s suggestion inspired us to further investigate this idea. Following the design of *Selective Kernel Networks* [4], we implemented a learnable kernel-selection variant of our DKA. Specifically, we construct two depthwise branches with different kernel sizes and fuse them using a lightweight attention gate: global average pooling → a two-layer MLP → softmax weights that dynamically select between the local and global branches for each input. This enables input-adaptive kernel aggregation. The results are reported in Table 3. While the learnable-kernel variant achieves competitive performance, our fixed 51×51 + 5×5 design remains slightly more robust in the low-data regimes (0.63% and 1.25%), likely because the learned gating requires additional data to generalize reliably. Under full data (100%), both approaches perform similarly.
> >
> > Overall, adaptive kernels are indeed feasible, but our chosen configuration offers greater robustness and reliability in the low-data medical scenarios that this work specifically targets.
> >
> > [4] Li, Xiang, et al. "Selective kernel networks." CVPR 2019.
> >
> > **Table 3: Comparison of learnable kernel sizes and our design on the COVID dataset using the pretrained ViT-B.**
> >
> > | **Methods**              | **0.63%** | **1.25%** | **100%** |
> > |--------------------------|-----------|-----------|----------|
> > | Learnable kernel sizes   | 88.98     | 91.05     | **99.21** |
> > | 51×51 + 5×5 (Ours)       | **89.01** | **91.06** | **99.21** |

---

> > > ### Author Response · Authors · 2025-11-22
> > > **Response to the reviewer dSfe (3/7)**
> > >
> > > ***Comment 6: Limited theoretical justification for why such large kernels are necessary.***
> > >
> > > **Reply:**
> > > Thank you for raising this important point. It is fair to note that our main goal is to provide a systematic empirical investigation of illustrating and mitigating the failures of  vanilla adapter in low-data regimes. In addition to the strong empirical results, we also provide extensive Fourier-domain (See Appendix F.9 in the revised submission) and ERF visualization (See Section 4.2 *ERF VISUALIZATION* in the revised submission) to illustrate why large kernels induce favourable inductive biases.
> > >
> > > To our best knowledge, most prior work on adapters [5-9] and large kernels [2,10] has been empirical rather than theoretical. A few recent studies explored the theoretical properties of convolutional architectures. For example, Caro et al.(2023) [11] show that small kernels tend to amplify high-frequency components, while larger kernels naturally suppress high-frequency variations. This observation aligns closely with our findings in Frequency-domain Analysis (See Appendix F.9 in the revised submission) that large-kernel adapters behave as smoother, low-pass residual correctors. In the medical imaging domain, this analysis is not only expected but desirable: anatomical structures tend to vary smoothly in space, while much of the acquisition noise, subtle artefacts, and spurious contrast fluctuations reside in higher frequencies. Medical images also often exhibit extensive anatomical variation and low contrast [12], which makes them vulnerable to overfitting on high-frequency noise when data are limited.
> > >
> > > However, we agree that fully characterizing the theoretical analysis of large-kernel adapters is an interesting and important direction. We plan to further strengthen the theoretical foundations of our approach in future work.
> > >
> > > [2] Liu, Shiwei, et al. "More convnets in the 2020s: Scaling up kernels beyond 51x51 using sparsity." ICLR (2022).
> > >
> > > [5] Houlsby, Neil, et al. "Parameter-efficient transfer learning for NLP." International conference on machine learning. PMLR, 2019.
> > >
> > > [6] Chen, Shoufa, et al. "Adaptformer: Adapting vision transformers for scalable visual recognition." Advances in Neural Information Processing Systems 35 (2022): 16664-16678.
> > >
> > > [7] Li, Jiashuo, et al. "Dynamic integration of task-specific adapters for class incremental learning." Proceedings of the Computer Vision and Pattern Recognition Conference. 2025.
> > >
> > > [8] Li, Yi-Chen, et al. "Q-Adapter: Customizing Pre-trained LLMs to New Preferences with Forgetting Mitigation." ICLR (2024).
> > >
> > > [9] Yang, Taojiannan, et al. "Aim: Adapting image models for efficient video action recognition." ICLR (2023).
> > >
> > > [10] Ding, Xiaohan, et al. "Scaling up your kernels to 31x31: Revisiting large kernel design in cnns." Proceedings of the IEEE/CVF conference on computer vision and pattern recognition. 2022.
> > >
> > > [11] Caro, Josue Ortega, et al. "Local convolutions cause an implicit bias towards high frequency adversarial examples." arXiv preprint arXiv:2006.11440 (2023).
> > >
> > > [12] Li, Hao, et al. "Large-kernel attention for 3D medical image segmentation." Cognitive Computation 16.4 (2024): 2063-2077.

---

> > > > ### Author Response · Authors · 2025-11-22
> > > > **Response to the reviewer dSfe (4/7)**
> > > >
> > > > ***Comment 7: Computational implications of such large kernels in 3D medical imaging are not discussed. Limited Applicability to Real Medical Imaging: The 51×51 kernel design is completely impractical for 3D medical imaging (MRI, CT scans). Computational Cost: What is the actual inference time and memory overhead for 51×51 convolutions on high-resolution medical images?***
> > > >
> > > > **Reply:**
> > > > We thank the reviewer for raising this point. We provide the following clarifications and analyses.
> > > >
> > > > **(1)** The proposed DKA is mainly designed for parameter-efficient fine-tuning in *2D* medical imaging settings, where low-data regimes make large-kernel adapters particularly beneficial. Three-dimensional volumetric segmentation presents markedly different computational and architectural considerations, necessitating specialised adapter designs. As our work focuses on 2D feature adaptation, extending the approach to 3D settings is beyond the current scope.
> > > >
> > > > **(2)** We agree that a naïve cubic 51×51×51 depthwise kernel would be prohibitively expensive. As shown in Table 4, such an isotropic 3D kernel incurs significantly higher FLOPs and memory compared with standard adapters, confirming that a direct 3D extension of our 2D design is impractical.
> > > >
> > > > **However**, following the work in [13] where they adopt anisotropic large-kernel forms in volumetric architectures, we employ a planar large kernel (i.e. 51×51×1) and found the required extra computational cost is manageable (as shown in below table).
> > > >
> > > > **(3)** Beyond the planar variant, multiple established strategies, such as kernel decomposition [14] and dilated 3D convolutions [15], offer promising and computationally efficient paths to generalise DKA to full 3D volumes. While developing such 3D-specific extensions is beyond our current scope, we believe our 2D findings motivate future exploration.
> > > >
> > > > [13] Liu, Siqi, et al. "3d anisotropic hybrid network: Transferring convolutional features from 2d images to 3d anisotropic volumes." International conference on medical image computing and computer-assisted intervention. Cham: Springer International Publishing, 2018.
> > > >
> > > > [14] Szegedy, Christian, et al. "Rethinking the inception architecture for computer vision." Proceedings of the IEEE conference on computer vision and pattern recognition. 2016.
> > > >
> > > > [15] Chen, Chen, et al. "3D dilated multi-fiber network for real-time brain tumor segmentation in MRI." International conference on medical image computing and computer-assisted intervention. Cham: Springer International Publishing, 2019.
> > > >
> > > > **Table 4: Computational cost of Adapter and DKA based on UNet3D.**
> > > >
> > > > | **Methods**          | **Inference Latency (ms)** | **Memory (MB)** |
> > > > |----------------------|----------------------------|-----------------|
> > > > | Adapter              | 12.44                      | 982.8           |
> > > > | DKA (51×51×1)        | 35.83                      | 1073.7          |
> > > > | DKA (51×51×51)       | 1139.06                    | 1257.3          |
> > > >
> > > >
> > > > ***Comment 8: Missing Comparisons: No comparison with recent vision-specific PEFT methods (e.g., Visual Prompt Tuning variants)***
> > > >
> > > > **Reply:**
> > > > Thank you for your valuable suggestion. Following your comment, we additionally compare our method with two recent vision-specific PEFT approaches, GateVPT [16] and DA-VPT [17], which represent advanced variants of Visual Prompt Tuning. As shown in Table 5, our method consistently outperforms both VPT-based baselines across all label ratios on the COVID dataset. These results further validate the effectiveness of our method compared with state-of-the-art prompt-based tuning methods.
> > > >
> > > > [16] Yoo, Seungryong, et al. "Improving visual prompt tuning for self-supervised vision transformers." International Conference on Machine Learning. PMLR, 2023
> > > >
> > > > [17] Ren, Li, et al. "DA-VPT: Semantic-Guided Visual Prompt Tuning for Vision Transformers." Proceedings of the Computer Vision and Pattern Recognition Conference. 2025.
> > > >
> > > > **Table 5: Comparison with Visual Prompt Tuning variants on the COVID dataset using the pretrained ViT-B.**
> > > >
> > > > | **Methods** | **0.63%** | **1.25%** | **100%** |
> > > > |-------------|-----------|-----------|----------|
> > > > | GateVPT [15] | 83.39     | 88.41     | 98.96    |
> > > > | DA-VPT [16]  | 84.85     | 89.19     | 99.02    |
> > > > | DKA       | **89.01** | **91.06** | **99.21** |

---

> > > > > ### Author Response · Authors · 2025-11-22
> > > > > **Response to the reviewer dSfe (5/7)**
> > > > >
> > > > > ***Comment 9: Limited exploration of other architectural modifications that could expand receptive fields. Alternative Architectures: Have you explored other ways to increase receptive fields, such as self-attention modules within adapters or hierarchical designs?***
> > > > >
> > > > > **Reply:**
> > > > > We clarify below that we have already explored a range of architectural variants in the manuscript and additionally provided more experiments to address your concern.
> > > > >
> > > > > **(1) Existing architectural modifications already explored.**
> > > > > As discussed in Figure 7 and Appendix Tables 13–16 in the revised submission, we have extensively evaluated alternative ways to enlarge receptive fields, including multiple dilated kernels, diverse kernel-size combinations, and different large-kernel configurations. These experiments already cover a set of architectural variants that influence effective receptive field.
> > > > >
> > > > > **(2) Additional architectures explored following the reviewer’s suggestion.**
> > > > > * (a) Self-attention within Adapter blocks.
> > > > > To the best of our knowledge, existing PEFT approaches do not introduce self-attention modules inside adapter branches. If the reviewer is aware of such designs, we would greatly appreciate these references.
> > > > >
> > > > > * (b) Hierarchical dilated designs.
> > > > > Motivated by the reviewer’s suggestion, we further implemented a hierarchical receptive-field expansion strategy by replacing our dual-branch structure with a three-layer 3×3 dilated stack using dilation rates \(\bar{d}=\{1,6,18\}\), yielding the same theoretical receptive field. As shown in Table 6, the hierarchical design performs competitively but remains consistently below our double-kernel branch, particularly in the low-data regimes. This suggests that our dual-kernel formulation offers a better balance between global context modeling and local detail preservation.
> > > > >
> > > > > Overall, while the reviewer’s suggestions are valuable and helped us broaden our evaluation, our proposed dual-kernel design remains the most effective and stable architecture for expanding receptive fields under strict data constraints.
> > > > >
> > > > > **Table 6: Comparison between a hierarchical dilated design and our large-kernel branch on the COVID classification dataset with the pretrained ViT-B. “\(\bar{d}\)” represents the dilation rate.**
> > > > >
> > > > > | **Methods**                                | **0.63%** | **1.25%** | **100%** |
> > > > > |--------------------------------------------|-----------|-----------|----------|
> > > > > | 3×3 Dilated Stack (\(\bar{d}=\{1,6,18\}\)) | 88.82     | 90.80     | 99.06    |
> > > > > | 51×51 + 5×5                                | **89.01** | **91.06** | **99.21** |
> > > > >
> > > > >
> > > > > ***Comment 10: Experimental Concerns: Batch size of 1 for segmentation might affect optimization dynamics.***
> > > > >
> > > > > **Reply:**
> > > > > Thank you for this comment. We address the concern from two complementary angles: **(i) clarifying why batch size = 1 was used in our original segmentation experiments**, and **(ii) providing new experiments with a larger batch size to verify that our conclusions remain unchanged**.
> > > > >
> > > > > **(1) Why batch size = 1 was used originally.**
> > > > > The use of batch size = 1 was due solely to hardware limitations: on our setup with NVIDIA RTX 3090 GPUs, any batch size larger than 1 resulted in out-of-memory errors. Importantly, to ensure strict fairness, all methods were evaluated under exactly the same configuration.
> > > > >
> > > > > **(2) Additional experiments with a larger batch size.**
> > > > > To address the reviewer’s concern, we further conducted additional segmentation experiments using a larger batch size of 4 with the same configuration. As shown in Table 7, increasing the batch size does not change the relative ranking: DKA continues to outperform Full Fine-tuning, Linear Probing, and Adapter across all label ratios. This confirms that our improvements are robust to batch-size variations and are not tied to a specific optimization setting.
> > > > >
> > > > > **Table 7: Comparison of methods on the BUSI segmentation dataset with batch Size = 4 based on the pretrained Segmenter-B.**
> > > > >
> > > > > | **Methods**        | **0.63%** | **1.25%** | **100%** |
> > > > > |--------------------|-----------|-----------|----------|
> > > > > | Full Fine-tuning   | 27.02     | 33.11     | 57.91    |
> > > > > | Linear Probing     | 25.81     | 32.47     | 54.60    |
> > > > > | Adapter            | 18.67     | 26.50     | 55.46    |
> > > > > | DKA                | **27.26** | **35.21** | **59.35** |
> > > > >
> > > > >
> > > > > ***Comment 11: The asynchronous learning rate finding (1e-3 for adapter, 1e-4 for head) seems important but is relegated to ablations.***
> > > > >
> > > > > **Reply:**
> > > > > Thank you for pointing this out. The asynchronous learning-rate setting is indeed an important part of our design. It was originally placed in the ablation section due to space constraints. In the revised submission, we have moved this configuration into the main experiment section (Section~4.4, *Asynchronous Learning Rates Matter*) and highlighted it in orange for clarity.

---

> ### Author Response · Authors · 2025-11-22
> **Response to the reviewer dSfe (6/7)**
>
> ***Comment 12: Dataset Size Discrepancy Makes Comparisons Misleading: Critical Issue: Figure 1 compares performance across datasets using percentages (0.63%, 1.25%, etc.) but these datasets have vastly different absolute sizes. For example, Tiny ImageNet has 100,000 training images while COVID has only 3,600. This means 0.63% of Tiny ImageNet (630 images) could be larger than 25% of COVID (~900 images). This makes the comparison fundamentally flawed - the paper is comparing different absolute data quantities while claiming to study "low-data" regimes. The authors should report absolute sample numbers and normalize comparisons appropriately.***
>
> **Reply:**
> Thank you for the comment. We respectfully disagree with the claim that our comparison setup is fundamentally flawed. We respond from two perspectives: **(i) why our comparisons are valid, and (ii) additional experiments using matched absolute sample counts**.
>
> **(1) Our comparisons remain valid and fair.**
> All baselines are compared *within the same dataset* under exactly the same number of training samples and settings. Across all datasets, DKA consistently achieves the best performance among all baselines. This stable superiority across datasets of very different sizes and characteristics clearly demonstrates the intrinsic strength of our method, rather than any artifact introduced by percentage-based sampling.
>
> **(2) Additional experiments under matched absolute sample sizes.**
> For Figure 1, to directly address the reviewer’s request, we further normalized datasets to the same absolute number of training samples. Since 0.63% and 1.25% of the COVID dataset correspond to approximately 120 and 240 samples, we applied these exact sample counts to Tiny-ImageNet and CIFAR-100, forming a fully controlled cross-dataset setting.
> As shown in Table 8, the conclusions remain unchanged. On *natural-image datasets*, the standard Adapter continues to outperform Linear Probing even at extremely small sample sizes—consistent with our paper’s finding. Meanwhile, DKA achieves the best performance.
>
> Overall, these results confirm that our method’s advantages do not depend on percentage-based sampling and remain robust under both percentage-matched and absolute-count-matched conditions.
>
> **Table 8: Comparison under equal absolute training sample sizes (120 and 240 samples) under the same configuration.**
>
> **Tiny-ImageNet**
>
> | Methods          | 120     | 240     |
> |------------------|---------|---------|
> | Linear Probing   | 6.54    | 10.38   |
> | Adapter          | 7.81    | 11.94   |
> | DKA              | **12.70** | **16.49** |
>
> **CIFAR-100**
>
> | Methods          | 120     | 240     |
> |------------------|---------|---------|
> | Linear Probing   | 21.22   | 30.79   |
> | Adapter          | 22.17   | 31.87   |
> | DKA              | **25.53** | **32.23** |
>
> **Covid**
>
> | Methods          | 120     | 240     |
> |------------------|---------|---------|
> | Linear Probing   | 86.84   | 87.50   |
> | Adapter          | 83.29   | 86.26   |
> | DKA              | **89.01** | **91.06** |

---

> ### Author Response · Authors · 2025-11-22
> **Response to the reviewer dSfe (7/7)**
>
> ***Comment 13: Missing Critical Implementation Details: LoRA Rank: The paper doesn't specify what rank was used for LoRA, which is crucial as rank fundamentally determines LoRA's capacity and parameter count. A rank-1 LoRA vs rank-64 LoRA are completely different methods. Without this information, the comparison with LoRA is meaningless.***
>
> **Reply:**
> Thank you for this comment. In all main-paper experiments, we used LoRA with rank \(r=4\), which is a standard and widely adopted configuration in prior PEFT works. To directly address the reviewer’s concern, we further evaluated LoRA with substantially larger ranks, including \(r=32\) and \(r=64\). The resulting trainable parameter counts and performance on the COVID dataset are reported in Table 9. Even with much larger ranks, corresponding to 1.2M and 2.4M additional trainable parameters, LoRA still underperforms our DKA across all data scales. These results confirm that DKA’s advantage is not due to a suboptimal LoRA configuration but persists even when LoRA is given significantly higher capacity.
>
> **Table 9: Comparison of LoRA with different ranks and DKA on the COVID dataset based on the pretrained ViT-B. TP represents trainable parameters.**
>
> | **Methods**     | **TP** | **0.63%** | **1.25%** | **100%** |
> |-----------------|--------|-----------|-----------|----------|
> | LoRA (r=32)     | 1.2M   | 84.21     | 86.90     | 99.00    |
> | LoRA (r=64)     | 2.4M   | 84.95     | 87.30     | 99.05    |
> | DKA             | 1.6M   | **89.01** | **91.06** | **99.21** |
>
>
> ***Comment 14: Unfair Parameter Comparison: While the paper claims DKA adds <2% parameters, there's no detailed comparison of parameter efficiency. The paper should provide: Exact parameter counts for each method; Performance vs. parameter trade-off curves; Comparison at iso-parameter budgets (e.g., what if LoRA used higher rank to match DKA's parameters?) Table in Figure 5 attempts this but only varies hidden dimensions, not comprehensively.***
>
> **Reply:**
> Thank you for your comment. Due to page limitations in the main paper, we only included Figure 5 as a compact illustration of parameter efficiency, without listing exact  parameter counts or performance.
>
> To fully address the reviewer’s request, we have added a comprehensive comparison across PEFT methods, reported in Table 10. For iso-parameter evaluation, we expanded LoRA to higher ranks, Adapter to larger hidden dimensions, and matched their resulting parameter ranges with the DKA setting. This ensures that all methods are compared under similar trainable-parameter budgets.
>
> As shown in Table 10, DKA not only achieves the highest accuracy among all methods but also delivers an order-of-magnitude higher efficiency ratio, far outperforming other baselines. These results confirm that DKA is not only competitive in raw accuracy but also substantially more parameter-efficient under both fixed-parameter and iso-parameter comparisons.
>
> **Table 10: Comparison of PEFT methods under matched trainable-parameter ranges. The table reports exact trainable parameters, accuracy, and the efficiency ratio ΔACC/ΔTP. d̂ is the hidden dimension in the adapter. r is the rank in the LoRA.**
>
> | **Methods**                         | **TP (M)**      | **ACC (%)**              | **ΔACC/ΔTP** |
> |-------------------------------------|-----------------|----------------------|--------------|
> | LoRA (r=4→64)                       | 0.2→2.4         | 63.64→66.34          | 1.23         |
> | Adapter (d̂=16→44)      | 0.6→1.6         | 63.18→64.09          | 0.91         |
> | Adapterformer (d̂=16→36) | 0.3→1.6         | 63.42→65.52          | 1.62         |
> | Convpass (d̂=16→29)     | 0.7→1.6         | 64.83→65.33          | 0.56         |
> | CIAT (d̂=16→87)         | 0.9→1.6         | 60.28→61.36          | 1.54         |
> | AIM (d̂=16→29)          | 0.9→1.6         | 62.72→63.61          | 1.27         |
> | DKA (11×11→51×51)                   | 0.7→1.6         | 65.81→74.23          | 9.36         |
>
>
> ---
>
> We appreciate your detailed comments and hope we have addressed your concerns. Please let us know if you have any additional questions.
>
> Sincerely,
> Authors

---

> > ### Comment · Reviewer_dSfe · 2025-11-26
> >
> > Thank you for your comprehensive and detailed responses to my concerns.
> > The authors have addressed the most critical experimental flaws (missing LoRA details, unfair parameter comparisons). The additional experiments strengthen the paper.
> > Given their comprehensive response, I am raising the score - they've fixed the methodological issues. The paper now presents a solid empirical study with proper experimental controls.
> > I appreciate the effort put into addressing the reviewers' concerns and believe these revisions have substantially improved the paper's rigor.

---

> > > ### Author Response · Authors · 2025-11-26
> > > **Thank you**
> > >
> > > We are delighted to hear that all your concerns have been thoroughly addressed. Your constructive feedback has been invaluable in improving the clarity of our work.  We sincerely appreciate your time and consideration in reviewing our submission and for raising the score.
> > >
> > > Warm regards,
> > >
> > > The Authors

---

### Author Response · Authors · 2025-12-04

Dear Area Chair,

Thank you for the time and effort dedicated to reviewing our work. We sincerely appreciate the constructive feedback from all reviewers, which has further strengthened our work and helped us clarify the novelty and contributions of our work more effectively. To support your evaluation, we provide below a clear and concise summary of our contributions.

- **Our contribution:**   First, we identify a previously unreported failure mode: standard adapters collapse under low-data medical settings. Our ERF-based diagnosis further shows that this degradation originates from a severely restricted receptive field. Second, we introduce DKA, a targeted, problem-driven solution derived directly from this analysis rather than a generic architectural extension. Overall, the contribution of our work is in discovering and addressing this overlooked failure mode, and the ERF-based perspective provides a new direction for future research on understanding and improving adapter behavior under data scarcity.

- Currently, three reviewers (@MhXB, @reej, @Mik3) have consistently given our submission a positive score of 6, recognizing both its contributions and technical soundness. Besides, we would like to highlight that the other reviewer @dSfe, who initially gave a score of 4, **subsequently raised their score to a positive 6 following our rebuttal, and this change occurred prior to the OpenReview identity leak event.**

We believe that our point-by-point responses below have thoroughly addressed all reviewer concerns. We thank you again for your time and consideration.

Warm regards,

The Authors

---

### Meta-Review · Area_Chair_fqoa · 2026-01-03

**Summary:**

Across the reviews, the primary concerns centered on limited theoretical depth, novelty relative to prior large-kernel adapter work, and practical considerations such as computational overhead and applicability to 3D medical imaging. Several reviewers noted that the core architectural idea—introducing large-kernel convolutions within adapters—appears conceptually simple and bears similarities to recent large-kernel adapter methods, raising questions about incremental novelty. Others requested stronger theoretical justification for the claimed causal relationship between effective receptive field (ERF) expansion and generalization, as well as clearer discussion of failure modes and limitations. Additional concerns included fairness of baseline comparisons, missing implementation details (e.g., LoRA rank, parameter efficiency), dataset-size normalization across low-data regimes, and the computational implications of very large kernels, particularly for volumetric medical imaging. These concerns were largely addressed through detailed rebuttal responses and additional experiments, leading all reviewers to converge to borderline positive scores, which ultimately supported acceptance.

**Reviewer Concerns:**

**Concerns addressed by the rebuttal:**

The rebuttal made a strong and convincing effort to address the majority of the reviewers’ concerns. In particular, the authors added extensive additional experiments and ablations that clarified the role of effective receptive field (ERF), demonstrated robustness across matched absolute sample sizes, and strengthened fairness of comparisons by providing missing implementation details (e.g., LoRA rank) and iso-parameter analyses. Reviewers’ concerns about baseline coverage were addressed through new comparisons with recent PEFT methods, and questions around optimization stability, batch size effects, and computational overhead were supported with concrete latency, FLOPs, and memory measurements. Several reviewers explicitly acknowledged that these additions resolved their main methodological and evaluation-related concerns, and at least one reviewer increased their score following the rebuttal.

**Concerns that remain outstanding:**

Some concerns remain only partially resolved. In particular, while the rebuttal provides stronger empirical and intuitive justification, the theoretical grounding of the ERF–generalization relationship is still largely heuristic rather than formally derived, and novelty relative to closely related large-kernel adapter methods remains somewhat incremental. In addition, although the authors clarified the scope and feasibility of extending the approach to 3D medical imaging, this limitation remains inherent to the current work. These remaining issues were generally viewed as scope and depth limitations rather than fundamental flaws, and did not outweigh the strengthened empirical evidence and clarity provided in the rebuttal.

Overall, the rebuttal substantially improved the paper, addressed the key technical concerns raised by reviewers, and reduced remaining issues to acceptable limitations, supporting the final decision to accept.

**Reviewer Scores:**

This paper initially received three borderline accept and one borderline reject. But after rebuttal, the reviewer who initially gave a negative score satisfied with the rebuttal and clearly stated to raise the score.

Therefore, after rebuttal, the paper received all positive scores even though all of them are borderline accept. AC does not have strong reason to reject the work apart from the incremental novelty.

---

### Decision · Program_Chairs · 2026-01-26

Accept (Poster)